# Effective Distillation to Hybrid xLSTM Architectures

**Lukas Hauzenberger** [1 2 *]  **Niklas Schmidinger** [1 2 *]  **Thomas Schmied** [1]  **Anamaria-Roberta Hartl** [1]  **David Stap** [2]
**Pieter-Jan Hoedt** [1]  **Sebastian Böck** [2]  **Günter Klambauer** [1]  **Sepp Hochreiter** [1 2]

## Abstract

There have been numerous attempts to distill quadratic attention-based large language models (LLMs) into sub-quadratic linearized architectures. However, despite extensive research, such distilled models often fail to match the performance of their teacher LLMs on various downstream tasks. We set out the goal of *lossless distillation*, which we define in terms of tolerance-corrected *Win-and-Tie rates* between student and teacher on sets of tasks. To this end, we introduce an effective distillation pipeline for xLSTM-based students. We propose an additional merging stage, where individually linearized experts are combined into a single model. We show the effectiveness of this pipeline by distilling base and instruction-tuned models from the Llama, Qwen, and Olmo families. In many settings, our xLSTM-based students recover most of the teacher's performance, and even exceed it on some downstream tasks. Our contributions are an important step towards more energy-efficient and cost-effective replacements for transformer-based LLMs.

## 1. Introduction

Current large language models (LLMs) require enormous computational resources due to their attention mechanisms (Vaswani et al., 2017; Touvron et al., 2023; Team, 2023; OpenAI, 2025), which scale quadratically with context length. As a result, these models are energy-intensive and costly to deploy. To address these limitations, many works aim to distill (Hinton et al., 2015) LLMs into linearized, attention-free, or more generally sub-quadratic architectures (Wang et al., 2024; Bick et al., 2024; Wang et al.,

---

*Equal contribution [1]Institute for Machine Learning, Johannes Kepler University, Linz, Austria [2]NXAI, Linz, Austria. Correspondence to: Lukas Hauzenberger <hauzenberger@ml.jku.at>, Niklas Schmidinger <schmidinger@ml.jku.at>.

*Proceedings of the 43rd International Conference on Machine Learning*, Seoul, South Korea. PMLR 306, 2026. Copyright 2026 by the author(s).

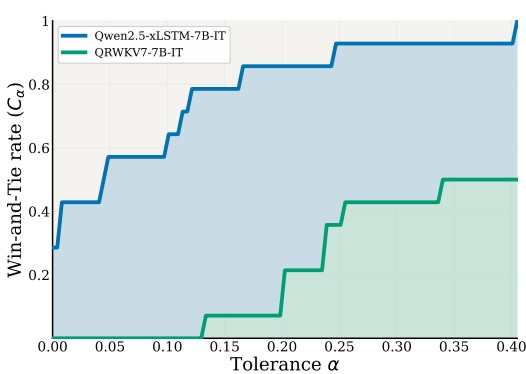

*Figure 1.* Win-and-Tie rate ($C_\alpha$) of our distilled QWEN2.5-7B-IT student and the best sub-quadratic baseline across generation benchmarks spanning math, code, STEM, and chat domains. Higher is better.

2025; Zhang et al., 2025; Lan et al., 2025; Goldstein et al., 2025). The efficient inference of sub-quadratic distilled LLMs makes them favorable drop-in replacements, if they match their teachers across a broad spectrum of tasks.

Recent post-training linearization has coalesced around a handful of sub-quadratic sequence mixer designs to *substitute* full softmax attention layers and a small set of recurring distillation techniques. LoLCATs (Zhang et al., 2025) and Liger (Lan et al., 2025) implement intra-layer hybrids that couple linear attention variants with sliding window attention (SWA), RADLADS (Goldstein et al., 2025) adapt RWKV-6 (Peng et al., 2024) and RWKV-7 (Peng et al., 2025) for the distillation setting, and Llamba (Bick et al., 2025) converts layers to Mamba-2 state-space mixers (Dao & Gu, 2024). For linearization, supervision typically involves hidden-state and logit alignment on small subsets of general web-text mixtures or instruction datasets. In contrast, token budgets for conventional LLM pre-training range from tens of billions to trillions of tokens, rendering linearization orders of magnitude more token-efficient than training from scratch. Therefore, linearization is an attractive fine-tuning regime for both exploring novel linear attention designs and lowering the deployment cost of Transformer-based models. However, existing linearization attempts have not yet achieved effective distillation. While linearized models often match the teacher on language understanding or knowledge benchmarks, they fall short on

harder generative evaluations that probe the student's mathematical reasoning or code synthesis abilities (see Figures 3b and 4). These outcomes highlight limitations of existing distillation procedures, architectures, and evaluation protocols (see Appendix B.2 for an overview of prior work).

**xLSTM as a powerful linear alternative for LLMs.** Recently, modern recurrent architectures, such as xLSTM (Beck et al., 2024) and Mamba (Gu & Dao, 2024), have emerged as competitive linear-complexity alternatives to Transformers in language (Beck et al., 2025b), computer vision (Alkin et al., 2025; Pöppel et al., 2025), biological modeling (Schmidinger et al., 2025), decision-making (Schmied et al., 2025), and time series (Auer et al., 2025). Concurrently, specialized kernels enable efficient chunkwise-parallel training for linear recurrent neural networks (RNNs) and xLSTM, substantially improving throughput on high-end accelerators (Beck et al., 2025a). Recent scaling-law analyses further indicate that xLSTM maintains competitive advantages as training and inference contexts grow, positioning it as a strong foundation for efficient long-context models (Beck et al., 2026). We hybridize xLSTM with sparse attention by combining an mLSTM with a SWA path and sink tokens using learned gates. Conceptually, this is related to recent attention hybrids that blend quadratic key-value (KV) memory with linear fast-weight memory (Irie et al., 2025). While our empirical validation focuses on mLSTM, the pipeline naturally extends to other sub-quadratic sequence mixers by replacing the global recurrent branch while keeping hidden-state alignment, sparse knowledge distillation, local SWA+sink attention, and optional expert merging unchanged.

**Contributions.** To rigorously assess whether linearized students can serve as drop-in replacements, we formalize a reliability criterion via the *Win-and-Tie rate* $C_\alpha$, which measures how broadly the student recovers teacher-level performance across benchmarks. Using this criterion, we show that prior linearization approaches often preserve language understanding but fall short on harder, free-form generation tasks. To close this gap, we introduce a linearization pipeline that replaces quadratic softmax attention with an efficient mLSTM–SWA hybrid. In our linearization pipeline we introduce a merging stage, where domain-specialized students are distilled independently and consolidated afterwards. In this sense, we demonstrate that linearization can be made modular: linearized models can be consolidated through simple weight-space merging (Wortsman et al., 2022). The resulting merge of distilled xLSTM students closes the performance gap on free-form generation tasks and consistently dominates existing linearization methods across tolerance levels on $C_\alpha$. While our empirical validation focuses on mLSTM, the pipeline naturally extends to other sub-quadratic sequence mixers by replacing the global recurrent branch while keeping hidden-state alignment, sparse knowledge

distillation, local SWA+sink attention, and optional expert merging unchanged.

**Conflict of Interest Statement.** Several authors were affiliated with NXAI GmbH during this work. NXAI GmbH develops xLSTM under an open license and commercial products based on xLSTM technology.

## 2. Background

**Softmax attention and Transformers.** The impressive capabilities of Transformer-based LLMs are largely attributed to the effectiveness of the underlying softmax-attention mechanism (Vaswani et al., 2017), which enables fine-grained modeling of long-range dependencies. At each time step $t$, an attention layer receives an input $\boldsymbol{x}_t \in \mathbb{R}^d$ and projects it to a query $\boldsymbol{q}_t$, key $\boldsymbol{k}_t$, and value $\boldsymbol{v}_t$ via learned linear maps $\boldsymbol{W}_q, \boldsymbol{W}_k \in \mathbb{R}^{d \times d_{qk}}$ and $\boldsymbol{W}_v \in \mathbb{R}^{d \times d_v}$:

$$\boldsymbol{q}_t = \boldsymbol{x}_t \boldsymbol{W}_q, \qquad \boldsymbol{k}_t = \boldsymbol{x}_t \boldsymbol{W}_k, \qquad \boldsymbol{v}_t = \boldsymbol{x}_t \boldsymbol{W}_v,$$

so that $\boldsymbol{q}_t, \boldsymbol{k}_t \in \mathbb{R}^{d_{qk}}$ and $\boldsymbol{v}_t \in \mathbb{R}^{d_v}$. To avoid recomputation, KV caches are maintained whose sizes grow with time, $\boldsymbol{K}_t \in \mathbb{R}^{t \times d_{qk}}$ and $\boldsymbol{V}_t \in \mathbb{R}^{t \times d_v}$, updated by concatenation (denoted as $[\,]$) along the time dimension:

$$\boldsymbol{K}_t = [\, \boldsymbol{K}_{t-1} \;\; \boldsymbol{k}_t \,], \qquad \boldsymbol{V}_t = [\, \boldsymbol{V}_{t-1} \;\; \boldsymbol{v}_t \,], \quad (1)$$

The output is then read from memory using *scaled softmax attention*:

$$\boldsymbol{h}_t = \mathrm{softmax}\left( \tfrac{1}{\sqrt{d_{qk}}} \boldsymbol{q}_t \boldsymbol{K}_t^\top \right) \boldsymbol{V}_t, \quad (2)$$

For a given query $\boldsymbol{q}_t$ at $t$, the dot product between the query and all stored keys $\boldsymbol{K}_t$ up to $t$ is computed. Subsequently, softmax is applied over time steps, and the resulting per-position attention scores are used to compute a weighted average of the stored values $\boldsymbol{V}_t$. During training and context encoding, each query in a sequence of length $T$ is compared with every key, incurring $O(T^2)$ time. During autoregressive inference, KV pairs are appended to the cache. At step $t$, the attention readout time and the cache size are $O(t)$. Although linear in $t$, the cache footprint scales with network depth and heads, and the cache must be read at every step, implying $O(t)$ memory-bandwidth cost. For long contexts, this becomes a dominant system bottleneck on modern accelerators, constraining batch size and throughput and increasing latency.

**Sparse and Sliding-window attention.** To mitigate the training and inference costs of full softmax attention, many LLMs adopt sparse attention patterns in which each head attends only to a subset of past positions (Child et al., 2019; Beltagy et al., 2020; Zaheer et al., 2020; Yuan et al., 2025). A widely used special case is sliding window attention (SWA), which restricts each query to attend to a

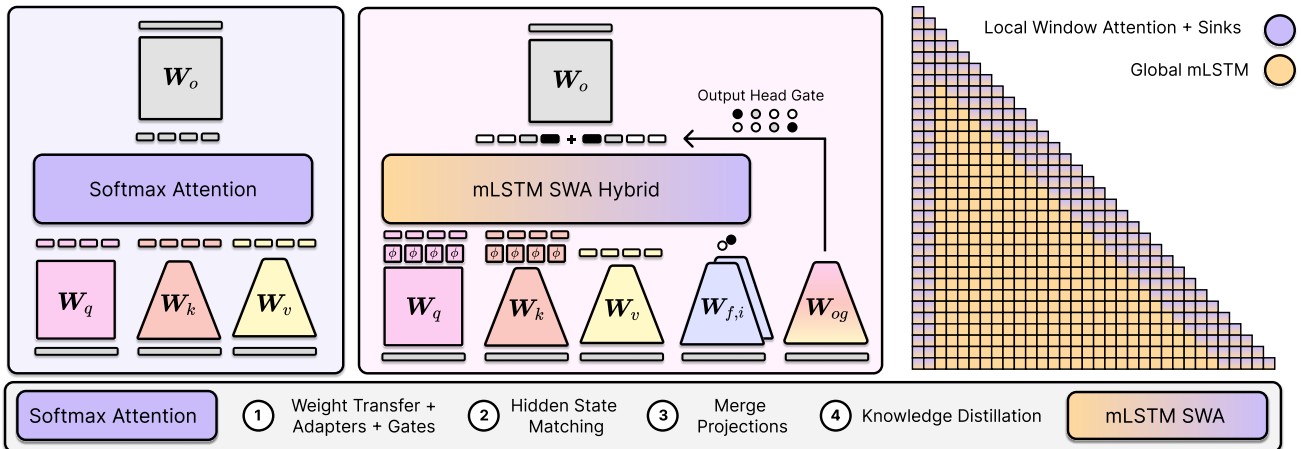

*Figure 2.* Illustration of our hybrid method consisting of mLSTM, sliding-window attention, and sink tokens. Our approach comprises 4 primary steps: **(1)** transfer the original teacher weights to the student and introduce adapters and gates, **(2)** hidden-state matching, **(3)** subsequent merging of query and key projections, and **(4)** knowledge distillation.

fixed-length band of its immediate token history. SWA evicts keys and values outside the last $W$ steps:

$$\boldsymbol{K}_t^W = \begin{bmatrix} \boldsymbol{K}_{t-1}^{W-1} & \boldsymbol{k}_t \end{bmatrix} \qquad \boldsymbol{V}_t^W = \begin{bmatrix} \boldsymbol{V}_{t-1}^{W-1} & \boldsymbol{v}_t \end{bmatrix},$$

where $\boldsymbol{K}_t^W = \boldsymbol{K}_{\max(1,t-W+1):t} \in \mathbb{R}^{\min(t,W) \times d_{qk}}$ and $\boldsymbol{V}_t^W = \boldsymbol{V}_{\max(1,t-W+1):t} \in \mathbb{R}^{\min(t,W) \times d_v}$. The maximum cache length of SWA therefore never exceeds $W$, while the core attention computation remains unchanged (see Equation 2). For sequences of length $T$, training and prefill of SWA can be implemented in linear $O(TW)$ time instead of $O(T^2)$ for full softmax attention. Consequently, during autoregressive decoding, both the computational and memory complexities of SWA are independent of the global sequence length. In Appendix C.1 we discuss the effective receptive field of SWA.

**Linear attention** replaces the exponential kernel of softmax attention $\kappa_{\exp}(\boldsymbol{q}, \boldsymbol{k}) = \exp\left(\boldsymbol{q}^\top \boldsymbol{k} / \sqrt{d_{qk}}\right)$ with a finite-dimensional feature map $\phi : \mathbb{R}^{d_{qk}} \to \mathbb{R}^{d_{qk}}$ such that $\kappa_\phi(\boldsymbol{q}, \boldsymbol{k}) = \phi(\boldsymbol{q})^\top \phi(\boldsymbol{k})$ (Katharopoulos et al., 2020). This factorization enables two efficient implementations of causal attention: a chunkwise-parallel form for training and context encoding and a strictly recurrent form for stepwise decoding (see e.g. Yang et al., 2024). Switching between these views enables prefill and training in linear time and constant-memory generation. In the recurrent view, we maintain a per-head KV state $\boldsymbol{S}_t \in \mathbb{R}^{d_{qk} \times d_v}$ that accumulates prefix statistics via rank-1 outer-product updates, together with an optional normalizer $\boldsymbol{z}_t \in \mathbb{R}^{d_{qk}}$:

$$\boldsymbol{S}_t = \boldsymbol{S}_{t-1} + \phi(\boldsymbol{k}_t) \otimes \boldsymbol{v}_t,$$
$$\boldsymbol{z}_t = \boldsymbol{z}_{t-1} + \phi(\boldsymbol{k}_t),$$

where $\otimes$ is the outer product. Given a query $\boldsymbol{q}_t$, we perform a normalized read from the current state:

$$\boldsymbol{h}_t = \frac{\phi(\boldsymbol{q}_t)\boldsymbol{S}_t}{\phi(\boldsymbol{q}_t)\boldsymbol{z}_t}.$$

Here, $\boldsymbol{q}_t, \boldsymbol{k}_t \in \mathbb{R}^{d_{qk}}$ and $\boldsymbol{v}_t \in \mathbb{R}^{d_v}$.

**mLSTM**. Inspired by the LSTM cell (Hochreiter & Schmidhuber, 1997), the mLSTM (Beck et al., 2024) augments the linear attention update with three data dependent gates that control distinct aspects of the update: $\boldsymbol{w}_i \in \mathbb{R}^{d \times 1}$, $\boldsymbol{w}_f \in \mathbb{R}^{d \times 1}$, $\boldsymbol{W}_{og} \in \mathbb{R}^{d \times d_v}$ where the input gate activations $i_t = \exp(\boldsymbol{x}_t \boldsymbol{w}_i)$ set the strength of the new KV write, the forget gate activations $f_t = \sigma(\boldsymbol{x}_t \boldsymbol{w}_f)$ decay the accumulated state, and the output gate activations $\boldsymbol{o}_t = \sigma(\boldsymbol{x}_t \boldsymbol{W}_{og})$ modulate the readout:

$$\boldsymbol{S}_t = f_t \, \boldsymbol{S}_{t-1} + i_t \, \phi(\boldsymbol{k}_t) \otimes \boldsymbol{v}_t,$$
$$\boldsymbol{z}_t = f_t \, \boldsymbol{z}_{t-1} + i_t \, \phi(\boldsymbol{k}_t),$$

Numerical stabilization for the exponential input gate is omitted for simplicity. A query then performs a normalized read, and the output gate modulates the retrieved value:

$$\hat{\boldsymbol{h}}_t = \boldsymbol{o}_t \odot \frac{\phi(\boldsymbol{q}_t)\boldsymbol{S}_t}{\phi(\boldsymbol{q}_t)\boldsymbol{z}_t}. \tag{3}$$

## 3. xLSTM distillation pipeline

In this work, we propose a distillation pipeline for creating efficient LLMs, substituting full softmax attention with a sub-quadratic attention proxy. The core of our method involves replacing the standard self-attention mechanism in a pre-trained LLM with a hybrid attention block that combines SWA with mLSTM[1] (Beck et al., 2024) via data-dependent gating.

---

[1] We use the xLSTM$_{[1:0]}$ configuration, which employs xLSTM blocks with mLSTM cells only.

### 3.1. Architecture & student initialization

We use a pretrained causal Transformer-based LLM as the *teacher* model, similar to prior work (Zhang et al., 2025). The *student* adopts the same high-level architecture design as the teacher, while replacing every multi-head attention block with a hybrid of SWA and mLSTM. This allows us to recycle the parameters of the original embedding and attention layers and the multi-layer perceptron (MLP) blocks. The fundamental motivation for our hybrid approach is to combine the strengths of two distinct and efficient sequence-mixing paradigms: the local context capturing ability of SWA and the linear complexity of mLSTM. Both components operate in parallel, and their outputs are dynamically fused using a learned, data-dependent gate.

**mLSTM adaptations**. Recent instantiations of gated linear operators replace the classical normalizer state with normalization layers such as LayerNorm (Sun et al., 2023; Yang et al., 2024; Beck et al., 2025a). In the linearization setting, we observe that adding normalization immediately before the output projections degrades student-teacher alignment. Similar observations have been made in (Bick et al., 2025). For this reason, we opt for the original normalizer design (cf. Equation 3) without normalization layers.

Instead of one output gate per channel, as in the original mLSTM, we use per-head scalar output gates to keep the parameter count closer to that of the teacher model. Furthermore, we found that using a concatenation of the head inputs over the feature dimension $\begin{bmatrix} \boldsymbol{q}_t & \boldsymbol{k}_t & \boldsymbol{v}_t \end{bmatrix}$ instead of the input activations $\boldsymbol{x}_t$ at time $t$ provides a better input signal for the output gate projections $\boldsymbol{W}_{og}$. Due to the strictly linear nature of the combination of $\boldsymbol{W}_q$, $\boldsymbol{W}_k$, $\boldsymbol{W}_v$ projections and $\boldsymbol{W}_{og}$, this can be merged to a single linear projection with input $\boldsymbol{x}_t$ at any stage. We augment the query and key inputs to the mLSTM with head-wise feature maps, applying softmax over the feature dimension as the activation function (Zhang et al., 2024).

**Attention Hybridization & Data-dependent Output Mixing.** We combine mLSTM and sparse attention into a single unified attention block, similar to Zhang et al. (2025) and Dong et al. (2025), rather than alternating both operators at every layer. We opt for a sparse attention pattern using SWA over the most recent token history and four initial tokens per sequence to preserve attention sinks, similar to Xiao et al. (2024). The combination of SWA and sink tokens enables both efficient KV cache compression and a good initial approximation of full softmax attention. For a discussion on attention sinks, we refer to Appendix C.2. In Section 4.3, we demonstrate that all three components are critical for strong performance. Moreover, in Appendix B.1, we contextualize our architectural design relative to contemporary hybrid linear-attention architectures.

For a given input batch, we compute query, key, and value activations and apply rotary position embeddings (RoPEs) (Su et al., 2024). The output of the *local* SWA + sink branch is computed using a sparse attention kernel (Dong et al., 2024). For the *global* mLSTM branch, we transform queries and keys with our head-wise feature maps $\phi$ and pass them together with input and forget gate activations to the mLSTM cell. Finally, the output gate produces a sigmoid-bounded scalar per head that modulates the global mLSTM against the local SWA + sink outputs, similar to Yuan et al. (2025) and Irie et al. (2025):

$$
\begin{aligned}
\hat{\boldsymbol{h}}_t &= o_t \, \mathrm{mLSTM}(\boldsymbol{q}_t) + (1 - o_t) \, \mathrm{SWA}(\boldsymbol{q}_t) \\
&= o_t \frac{\phi(\boldsymbol{q}_t)\boldsymbol{S}_t}{\phi(\boldsymbol{q}_t)\boldsymbol{z}_t} + (1 - o_t) \, \mathrm{sm}\!\left( \frac{\boldsymbol{q}_t \boldsymbol{K}_t^{W\top}}{\sqrt{d_{qk}}} \right) \boldsymbol{V}_t^W,
\end{aligned} \quad (4)
$$

where sm is used as a short form for softmax. This simple yet effective combination of mLSTM and SWA yields a harmonic interplay between modeling short and long-term dependencies.

**Extension to other sub-quadratic sequence mixers.** Although we instantiate the global branch with an mLSTM, the distillation pipeline is not tied to this particular recurrent cell. The essential requirement is that the replacement operator exposes a sub-quadratic causal sequence-mixing path that can consume projections derived from the teacher attention block and produce a per-token output in the same hidden space as softmax attention. Many recent efficient architectures satisfy this template, including gated linear attention, RetNet-style retention mechanisms, Mamba-like state-space mixers, Gated DeltaNet variants, and RWKV-style recurrent mixers. In such cases, the mLSTM term in Equation (4) can be replaced by a generic sub-quadratic mixer

$$
\mathrm{Mixer}_\psi(\boldsymbol{q}_t, \boldsymbol{k}_t, \boldsymbol{v}_t, \boldsymbol{s}_{t-1}),
$$

with architecture-specific recurrent state $\boldsymbol{s}_{t-1}$ and parameters $\psi$, while retaining the local SWA+sink branch and the data-dependent output mixing gate:

$$
\hat{\boldsymbol{h}}_t = o_t \, \mathrm{Mixer}_\psi(\boldsymbol{q}_t, \boldsymbol{k}_t, \boldsymbol{v}_t, \boldsymbol{s}_{t-1}) + (1 - o_t) \, \mathrm{SWA}(\boldsymbol{q}_t).
$$

The subsequent training recipe remains unchanged: Stage I aligns the replacement operator to the teacher attention outputs, Stage II performs end-to-end sparse knowledge distillation, and Stage III optionally merges independently distilled experts in weight space. Thus, our contribution should be viewed as a post-training linearization recipe for hybrid sub-quadratic students, with mLSTM serving as the concrete and empirically validated instantiation in this work.

### 3.2. Linearization fine-tuning

**Linearization Stage I: layer-wise hidden-state alignment.** Following prior linearization work (see Appendix B.2), we

first align the per-layer representations of the student to the attention outputs of the teacher using a mean-squared error (MSE) objective. For each layer $\ell$ and time step $t$, let $\boldsymbol{h}_t^{(\ell)} = \text{SoftmaxAttention}(\boldsymbol{q}_t^{(\ell)}, \boldsymbol{K}_t^{(\ell)}, \boldsymbol{V}_t^{(\ell)})$ denote the teacher's attention output and let $\hat{\boldsymbol{h}}_t^{(\ell)}$ denote the corresponding student hidden state as defined in Equation (4). The layer-wise objective is:

$$\min_{\boldsymbol{\theta}_\ell} \left\| \boldsymbol{h}_t^{(\ell)} - \hat{\boldsymbol{h}}_t^{(\ell)} \right\|_2^2, \tag{5}$$

where $\boldsymbol{\theta}_\ell$ denotes the newly introduced parameters, i.e., the parameters of the head-wise feature maps and gate projections. The embedding and MLP weights from the teacher are frozen in this stage. The full batch loss is then computed as the sum of Equation (5) over layers and time.

**Linearization Stage II: sparse knowledge distillation.** Following the hidden-state alignment stage, we unfreeze all student parameters $\boldsymbol{\theta}$ and fine-tune end-to-end. The objective for this stage interpolates between next-token prediction and matching the teacher distribution via the Kullback-Leibler divergence (KL):

$$\min_{\boldsymbol{\theta}} \Big\{ -\sum_{t=1}^{T} \gamma \, \log p_{\boldsymbol{\theta}}\left(y_t \mid \mathbf{x}_{1:t}\right)$$
$$+ \beta \, \text{KL}\left[ p_T^{(k)}\left(\cdot \mid \mathbf{x}_{1:t}\right) \,\big\|\, p_{\boldsymbol{\theta}}^{(k)}\left(\cdot \mid \mathbf{x}_{1:t}\right) \right] \Big\}, \tag{6}$$

where $p_T(\cdot \mid \cdot)$ and $p_{\boldsymbol{\theta}}(\cdot \mid \cdot)$ denote the teacher and student distributions, respectively. The superscript $(k)$ denotes the distribution over the top-$k$ tokens, giving rise to a sparse KL. For our experiments, we set $k = 256$ (cf. Team, 2025a). The sparse KL in Equation (6) makes it possible to precompute and store teacher targets over the full distillation dataset. As a result, the teacher does not need to be accessed directly during stage II. This is especially advantageous for long-context distillation, where querying an online teacher can become prohibitively costly. Scaling this regime efficiently will be an important focus of future work.

**Optional Stage III: Expert Merging.** Stages I–II can be applied either in a multi-task setting (one generalist student) or in a *decentralized* setting where $K$ domain experts (e.g., math, code, STEM, etc.) are trained in parallel, all starting from the same initialized seed weights $\boldsymbol{\theta}^{(0)}$. This branch-train-merge workflow mirrors a broader trend in post-training pipelines that target specific capabilities and later consolidate them into a single deployable model (DeepSeek-AI, 2025; Cohere, 2025; Team, 2026). Concretely, after distilling linear experts $\{\boldsymbol{\theta}^{(i)}\}_{i=1}^{K}$, we form a single student via simple linear weight merging (Wortsman et al., 2022):

$$\boldsymbol{\theta}_{\text{merge}} = \sum_{i=1}^{K} \lambda_i \, \boldsymbol{\theta}^{(i)}, \qquad \lambda_i \geq 0, \; \sum_{i=1}^{K} \lambda_i = 1, \tag{7}$$

with uniform weights by default and optional validation-tuned $\lambda_i$ when emphasizing particular capabilities. In our setting, this enables *capability patching*: researchers can independently improve a specific domain expert and update the final hybrid student by re-merging, without re-training the full model end-to-end. Moreover, the expert-centric setup is particularly well-suited for applying domain-specific fine-tuning or on-policy distillation to each expert before merging, i.e., learning from self-generated trajectories with teacher feedback, which we leave for future work (Agarwal et al., 2024). For a brief overview of decentralized post-training pipelines and model merging, see Section B.3.

## 4. Experiments

In this section, we apply our linearization protocol to both base models and instruction-tuned models from the Llama, Qwen, and Olmo families. We conduct downstream evaluations of the resulting hybrid models on established benchmarks across two important domains: **(1)** language understanding & knowledge tasks, and **(2)** language generation & reasoning tasks. Across benchmarks, we compare our distilled xLSTM students both against their teacher models and state-of-the-art linearization alternatives, including LoLCATs (Zhang et al., 2025), RADLADS (Goldstein et al., 2025), and Mamba-in-Llama (Wang et al., 2024). We leverage `lm-eval` (Gao et al., 2024) for conducting our evaluations (see Appendix E.3 for details). For mathematical evaluations, we use the `Math-Verify` package.

**Baseline comparability.** We evaluate prior linearization baselines using their publicly released checkpoints and original training recipes. Consequently, these comparisons are not fully controlled for data mixture, token budget, teacher model, number of distillation stages, optimization strategy, or architectural constraints such as retained attention layers. For example, some baselines use larger teachers or additional post-training stages, while others use smaller distillation budgets or different data mixtures. We therefore interpret these results as a comparison of available end-to-end linearization recipes rather than an isolated, compute-matched ablation of individual architectural choices.

**Metrics for effective distillation: teacher-recovery rate and tolerance-corrected win-and-tie rate.** Similar to Goldstein et al. (2025), we report the respective *teacher-recovery rate* as a primary per-benchmark metric, defined as the ratio between student and teacher performance. A recovery rate $> 1$ indicates that the student exceeds its teacher on the respective benchmark. Such cases should be interpreted as benchmark-specific gains under the full distillation and continued-training recipe, not as evidence that the student uniformly dominates the teacher. We refer to Appendix Section E.3 for absolute scores. However, when comparing distilled models across a diverse suite of benchmarks,

recovery rates alone do not quantify whether a student is a *reliable* drop-in replacement. In particular, simple aggregates of recovery (e.g., mean/median recovery) can obscure substantial regressions on a subset of tasks, and ratio-based summaries can be uninformative when the teacher scores are small, yielding misleadingly large or noisy relative changes. We therefore complement recovery rates with a *tolerance-corrected win-and-tie* metric that summarizes *task-level win-rate* across benchmarks. Following our definition of (approximately) lossless distillation (Appendix Section A), we compute the win-and-tie rate $C_\alpha$, i.e., the fraction of benchmarks on which the student matches or exceeds teacher performance within a tolerance $\alpha$. This metric captures parity coverage across heterogeneous evaluations and distinguishes truly lossless distillation from partial recovery. For compact model comparison, we report $\alpha^*$: the minimum tolerance $\alpha$ such that $C_\alpha \geq 0.5$. Lower $\alpha^*$ indicates a better student, and thus a better distillation process, since less tolerance is required to match the teacher on half of the benchmarks.

### 4.1. Base Model Evaluation: Validating the Hybrid Architecture

To assess the generality of our linearization pipeline for base models, we distill both LLAMA3.1-8B and OLMO3-7B. Olmo's fully open pre-training corpus provides a unique opportunity to evaluate whether matching the teacher on the original data distribution improves distillation compared to using alternative public datasets.

**Experimental setup**. For both models, we conduct stage I hidden-state alignment over 655M tokens with a sequence length of 4K using a standard linear-warmup to peak learning rate of $10^{-2}$ and cosine decay to $10^{-5}$. For Llama we leverage the Dolmino dataset[2], and for Olmo we use the Dolmino 3 midtraining mix[3] released as part of OLMO2 (OLMo, 2025) and OLMO3 (Olmo, 2025) and maintain the originally proposed mixing weights.

For stage II, we further distill our aligned Llama checkpoints on an additional 5 billion tokens from the same data mixes and context size as in phase I. For Olmo, we extend the token budget to 20 billion tokens to align the budget with the protocol used for instruction-tuned models (cf. Section 4.2). For both models, we train using $\gamma = 0.9$ and $\beta = 0.1$ for cross-entropy (CE) and KL losses, respectively, and rewarm to a constant learning rate of $10^{-5}$. Moreover, we provide additional experiment details, including a description of training settings and hyperparameters in Appendix D.

---

[2] https://huggingface.co/datasets/allenai/dolmino-mix-1124

[3] https://huggingface.co/datasets/allenai/dolma3_dolmino_mix-100B-1025

**Results**. First, we evaluate our xLSTM-based students on six established multiple-choice (MC) and log-likelihood tasks, such as MMLU (Hendrycks et al., 2021), that test for general language understanding and knowledge. Among publicly available baselines, LOLCATS is distilled from the same LLAMA3.1-8B teacher, enabling a direct recovery-rate comparison, while QRWKV6-7B is distilled from a Qwen-family teacher (Goldstein et al., 2025). We observe that our distilled students achieve full (xLSTM-LLAMA3.1-8B) or near-full (xLSTM-OLMO3-7B) teacher parity, while LOLCATS and QRWKV6-7B exhibit a significant performance gap. We report the respective teacher-recovery rates in Figure 3a. Additionally, absolute scores are reported in Table 5. Next, we evaluate our distilled models on a broad battery of commonly used language generation and reasoning tasks that span important domains such as mathematics and coding (Cobbe et al., 2021; Austin et al., 2021). Unlike language understanding tasks, these benchmarks test the model's ability to produce consistent and relevant answers. In Figure 3b, we report the recovery rate of our xLSTM-based students and established baselines (see Table 6 for the raw scores). We discover that prior methods exhibit significant performance gaps compared to the teacher, with LOLCATS and QRWKV6-7B both yielding $\alpha^\star = 1.0$. In contrast, our hybrid models achieve strong relative scores across most tasks, achieving $\alpha^\star = 0.0$ for LLAMA3.1-8B and $\alpha^\star = 0.01$ for OLMO3-7B.

**Interpreting student-over-teacher gains.** Several benchmarks show recovery rates above one, i.e., the distilled student exceeds its teacher on that benchmark. We do not interpret these gains as evidence that linearization alone creates capabilities absent from the teacher. Rather, the student is optimized with a mixed CE/KL objective on high-quality mid-training data, so stage II acts as distillation and continued training. This effect is most visible for xLSTM-LLAMA3.1-8B, which is distilled on Dolmino data that was not part of the original Llama 3.1 training mixture to our knowledge. By contrast, xLSTM-OLMO3-7B is distilled on the Dolmino 3 mid-training mixture of the OLMO family and does not show large student-over-teacher gains, suggesting that data distribution and continued training substantially contribute to improvements. Moreover, recovery rates can exaggerate relative gains when the teacher score is low, so we report absolute scores in Appendix E.3 and rely on $C_\alpha$ as a complementary parity measure.

### 4.2. Instruction-Tuned Model Evaluation: Validating Decentralized Linearization

**Experimental setup**. Next, we apply our linearization pipeline to *post-trained* models, focusing on LLAMA3.1-8B-IT and QWEN2.5-7B-IT. To ensure coverage of the capabilities of interest, we use our decentralized linearization scheme: starting from the teacher weights, we train

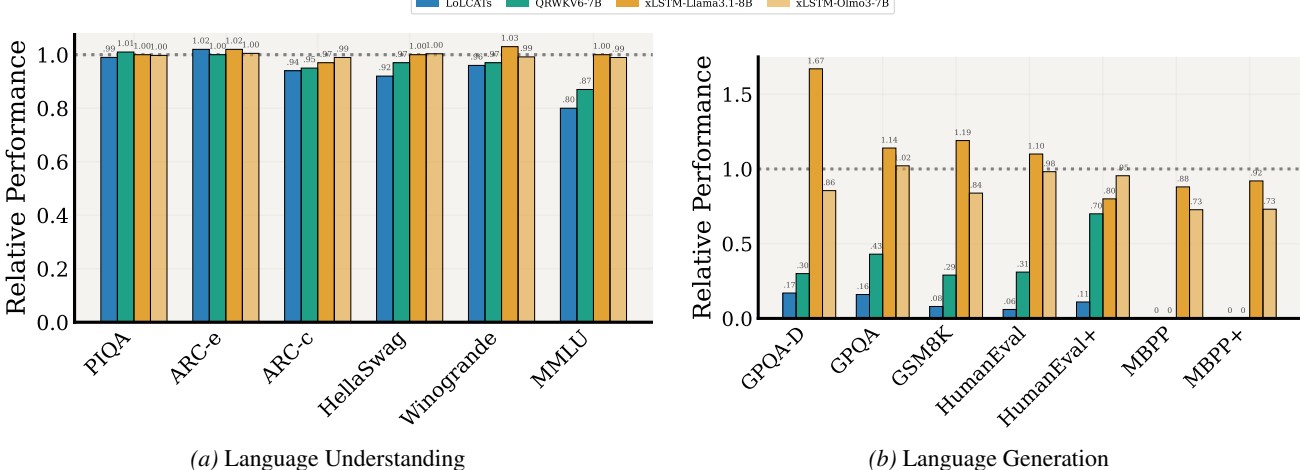

*(a)* Language Understanding                    *(b)* Language Generation

*Figure 3.* **Downstream evaluations** for **(a)** language understanding and **(b)** language generation tasks. We report the recovery rate relative to teacher scores for our mLSTM-based student and established baselines with comparable parameter counts. The dotted line at 1.0 indicates parity with the Transformer teacher. Our model matches the teacher's performance across language understanding tasks, while exceeding the teacher on four of the considered generation tasks.

four linearized specialists targeting **math**, **STEM**, **code**, and **instruction-following/chat**. Each expert is trained for ~5B tokens on a domain-specific mixture constructed from Nemotron Nano-2/3 and OLMO-3 data (NVIDIA, 2025b;c; Olmo, 2025; full mixtures and sampling weights in Table 2). As for base models, we use the Stage II objective in Eq. (6) with $\gamma = 0.9$ and $\beta = 0.1$, and train with a constant learning rate of $7 \times 10^{-7}$ for QWEN and $10^{-5}$ for LLAMA. After training, we consolidate the specialists into a single deployable student via linear weight merging (Eq. (7)). We tune the merge weights $\lambda_i$ on held-out validation splits where available (GSM8K, MATH, MBPP), otherwise on held-out distillation data via validation perplexity and simple heuristics (e.g., downweighting experts that underperform across domains). Test-only benchmarks (HumanEval, CruxEval, GPQA, IFEval, MT-Bench) are not used for selection, so their recovery rates reflect generalization rather than overfitting (full protocol in Appendix D.3).

**Results.** Beyond language understanding benchmarks, we evaluate both the individual experts and the merged student on an expanded suite of generative benchmarks spanning our target domains (math, STEM reasoning, code, and instruction-following). Figure 4 reports teacher-recovery rates for each benchmark and highlights the effect of merging by comparing the merged checkpoint with its constituent specialists.

We compare against instruction-tuned linearization baselines with aligned teachers. For LLAMA3.1-8B-IT, we use MAMBA-IN-LLAMA, a Mamba2 attention hybrid that retains 50% of the original softmax attention layers. For QWEN2.5-7B-IT, we compare to QRWKV7-7B-IT (Goldstein et al., 2025). We find that our decentralized distillation pipeline yields strong linearized students that match their

teachers on language-understanding benchmarks and code-generation evaluations, while recovering most of the teacher performance on mathematical reasoning tasks. We additionally assess instruction-following quality on MT-bench (Zheng et al., 2023) using LLM-as-a-judge, where GPT-5.1 grades generated responses. Across both the LLAMA and QWEN families, our students receive higher preference scores than their respective teachers (see Appendix E Figure 9 for detailed results). The largest remaining gap appears in STEM reasoning, where the merged student underperforms the dedicated STEM expert. Overall our merged students exhibit strong performance with xLSTM-LLAMA3.1-8B-IT and xLSTM-QWEN2.5-7B-IT achieving $\alpha^\star = 0.02$ and $\alpha^\star = 0.05$ respectively. We therefore conclude that expert training and subsequent model merging is a promising strategy to distill capabilities of interest in parallel and unify linearized models.

**Model merging discussion** Across both model families, we observe *positive transfer* when unifying independently trained experts into a single hybrid student. Most notably, merging substantially improves instruction-following for Llama, where we recover a large fraction of the gap on IFE-VAL compared to the instruction expert alone. At the same time, merging is not uniformly beneficial. Both students exhibit the most pronounced degradations on STEM-oriented evaluations (e.g., GPQA and GPQA-DIAMOND), indicating interference between domain updates. In contrast, math and code capabilities are largely robust to merging, and for QWEN2.5-7B-IT we also observe comparatively minor changes in instruction-following. Overall, these results validate that simple weight-space merging can be effective even for *fully linearized* architectures, opening a path toward consolidating independently developed efficient students.

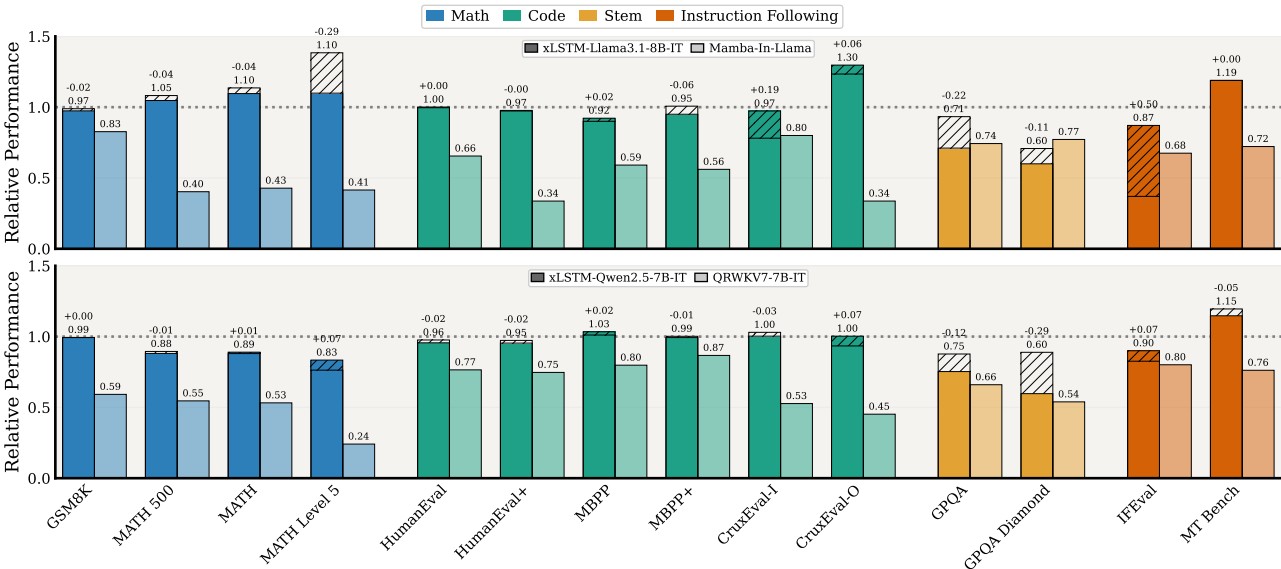

*Figure 4.* **Teacher-recovery rates for instruction-tuned xLSTM students and the effect of expert merging. Top:** xLSTM-LLAMA3.1-8B-IT distilled from LLAMA3.1-8B-IT vs. MAMBA-IN-LLAMA; **Bottom:** xLSTM-QWEN2.5-7B-IT distilled from QWEN2.5-7B-IT vs. QRWKV7-7B-IT. For each benchmark (x-axis; grouped by domain color), we report *relative performance* as the student/teacher score ratio (y-axis); the dotted line at 1.0 indicates parity with the Transformer teacher. For our method (left bar in each pair), the *merged* student is shown. For a given task, the striped area on top of the bar indicates gains (colored) or losses (empty) compared to our linearized *domain expert* before merging. For the baselines (right bar in each pair), the light bar shows the recovery rate.

We use simple linear averaging because it is strong, cheap, and practical. A modified TIES-style baseline (Yadav et al., 2023) applies cleanly only to parameters shared with the teacher, since the introduced mLSTM and SWA parameters have no teacher/base-model counterpart for the task-vector view ($\Delta^{(i)} = \boldsymbol{\theta}^{(i)} - \boldsymbol{\theta}^{(0)}$); in three-way TIES experiments restricted to these shared parameters, performance did not exceed linear averaging. We therefore treat weight-space merging as an effective practical recipe rather than a theoretically settled solution to multi-task interference.

**Synthetic long context limitations**. In Appendix Table 8, we evaluate Needle in a Haystack accuracy for single needle and multi needle retrieval across 1k, 4k, 8k, and 16k tokens (Hsieh et al., 2024). The instruction tuned Transformer baselines LLAMA3.1-8B-IT and QWEN2.5-7B-IT retain near perfect recall across all tested lengths, whereas our distilled students exhibit degradations with increasing context length, with the most pronounced decline starting at the training context size of 4k tokens. This identifies long context retrieval as a limitation of the current distillation recipe, but does not imply an inherent limitation of hybrid recurrent architectures. PRISM shows that post training can also substantially degrade long context retrieval in pure Transformers, and that weight merging combined with targeted recovery training can partially restore this capability (Runwal et al., 2026). Moreover, recent hybrid models such as Nemotron, Qwen3 Next, and SWAX demonstrate that architectures combining linear or fixed size memory with attention can achieve strong long context performance (NVIDIA, 2025c; Team,

2025c; Cabannes et al., 2025). Future work could address this limitation by training dedicated long-context specialists within our decentralized linearization recipe or by retaining a small number of full-attention layers, whereas this work deliberately studies the limits of fully linearized students.

### 4.3. Ablations

**Setup.** For the following ablations we distill linear student models from LLAMA3.1-8B with a token budget of 2.5 billion tokens.

**Effect of Distillation Objective.** The distillation objective is another critical component of every distillation recipe. Therefore, we conduct an ablation in which we vary the mixture weights $\gamma$ and $\beta$ for the CE and KL losses used in stage II, respectively. We observe that $\gamma = 0.9$ and $\beta = 0.1$ result in a low cross-entropy loss, while preventing the student from drifting too far away from its teacher (see Appendix F.2). In contrast, training purely via KL resulted in worse performance compared to a mixed objective due to overconstraining the linear student to the teacher. This trend persists even when distilling on the teacher's original training data, indicating that the gains are not merely an artifact of *up-training* the student on higher-quality data.

**Effect of mLSTM, SWA & Sinks.** Our hybrid approach utilizes a combination of mLSTM, SWA with a fixed size of 512 tokens and 4 sink tokens. In Figure 15, we empirically validate the individual contributions of these components.

In addition, we compare against a pure linear attention baseline to understand the contribution of mLSTM. Pure mLSTM exhibits a considerably lower loss than linear attention, which highlights the effectiveness of its gating mechanism. The combination of mLSTM and SWA results in a striking improvement, pointing towards a harmonic relationship between modeling short and long-term dependencies via SWA and mLSTM, respectively. We report additional analyses on the importance of sink tokens and SWA in Appendix F.1. Moreover, we analyze the mixture weights produced by the data-dependent output gate and find that both contribute considerably to the final outputs (see Appendix E.2).

**PEFT vs. FFT.** Prior linearization recipes use parameter-efficient fine-tuning (PEFT) via low-rank adaptation (LoRA, Hu et al., 2022) to recover performance lost during conversion (Zhang et al., 2025; Nguyen et al., 2025). While LoRA is cheaper, it is also less expressive than full fine-tuning (FFT). We find that FFT results in considerably improved performance and therefore adopt it by default in our distillation recipe (see Appendix F.3).

### 4.4. Inference Comparison

A key motivation for distilling Transformer teachers into recurrent xLSTM students is improved serving efficiency. Following prior work (De et al., 2024; Beck et al., 2025b), we report inference results separately for *prefill* (prompt encoding) and *generation* (autoregressive decoding) (Pope et al., 2023). Our inference tests are characterized by three hyperparameters: batch size $B$, context length $C$, and generation budget $G$. Teacher and student are implemented in `transformers` (Wolf et al., 2020) and optimized with `torch.compile`, FlashAttention (Dao, 2024), and fused mLSTM kernels (Beck et al., 2025a). The student uses a static cache storing mLSTM states plus SWA KV (see Appendix E.4).

**Prefill.** Figure 10 reports throughput and time to first token (TTFT) on one H100 80GB for the largest feasible $B/C$ pairs. Our hybrid student is consistently faster, with $\sim 2\times$ higher throughput at $B{=}1$, $C{=}65$K and an overall $\sim 2\times$ reduction in TTFT.

**Generation.** Figures 5 and 11 compare decoding latency, memory, and throughput. Without prefill (to isolate decoding complexity), the student roughly halves latency and GPU memory at $G{=}131$K, while maintaining constant memory over time. With prefill and $B{=}8$, the student achieves up to $\sim 4\times$ higher throughput as context length grows. The teacher runs out of memory at larger batches.

We demonstrate strong efficiency benefits of our xLSTM-based student in terms of latency, throughput, and memory consumption. We note that other linearized methods, such as LoLCATs (Zhang et al., 2025), exhibit similar inference

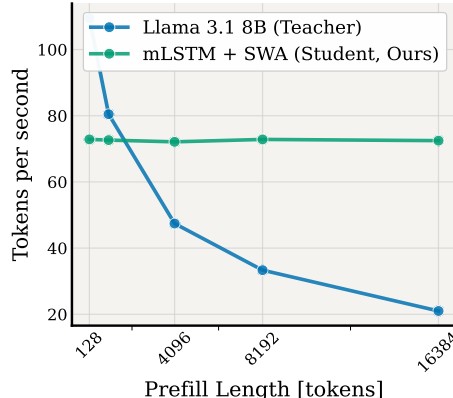

*Figure 5.* Generation Throughput of LLAMA3.1-8B (blue) and xLSTM-LLAMA3.1-8B (green) for batch size 8.

advantages (but with a larger student-teacher gap on downstream tasks) and therefore omit an additional comparison.

### 5. Conclusion

We presented a linearization pipeline that replaces quadratic softmax attention with an efficient mLSTM–SWA hybrid. To assess whether distilled students are reliable drop-in replacements, we formalized lossless distillation through the *Win-and-Tie rate* $C_\alpha$: the fraction of benchmarks on which the student matches or exceeds teacher performance (optionally within tolerance $\alpha$), and its critical tolerance $\alpha^*$, the minimum $\alpha$ such that $C_\alpha \geq 0.5$, i.e., the tolerance needed to match the teacher on at least half of benchmarks. Across base and instruction-tuned teachers from the Llama, Qwen, and Olmo families, our xLSTM students attain substantially higher $C_\alpha$ and Pareto-dominate prior linearization methods across tolerances, reflecting more effective distillation across benchmarks rather than gains on a small subset. As a result, our hybrid student models are prime candidates for a drop-in replacement of full-attention transformers when inference efficiency matters. We further showed that *distilling domain experts* and consolidating them via simple weight-space merging improves $C_\alpha$, indicating that weight merging remains effective after full linearization and enabling modular capability development and targeted updates.

**Limitations and future work.** The remaining deficits are most visible on synthetic long-context evaluations and on STEM-oriented benchmarks (e.g., GPQA, GPQA-DIAMOND), where interference between independently trained experts can reduce recovery after merging. A key next step is to further probe which expert domains are most beneficial to distill in isolation and how to consolidate them more reliably. For long-context behavior, we plan to explore stronger attention hybrids and memory designs. Finally, we aim to scale this recipe to larger teachers, including Sparse-Mixture-of-Experts models, and to study on-policy distillation or RL-based expert refinement prior to merging.

## Acknowledgments

This work was supported by European Union's Horizon Europe research and innovation programme under grant agreement number 101214398 (ELLIOT) and the Austrian Science Fund (FWF) 10.55776/COE12. The ELLIS Unit Linz, the LIT AI Lab, the Institute for Machine Learning, are supported by the Federal State Upper Austria. We acknowledge EuroHPC Joint Undertaking for awarding us access to Leonardo at CINECA, Italy, Deucalion at MACC, Portugal, and Discoverer at SofiaTech, Bulgaria.

## Impact Statement

This paper presents work whose goal is to advance the field of Machine Learning. There are many potential societal consequences of our work, none which we feel must be specifically highlighted here.

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

# A. Definition of Knowledge Distillation Goals

> **Lossless and $\alpha$-Tolerant Knowledge Distillation**
>
> *Knowledge distillation* (KD) aims to transfer the knowledge of a pre-trained *teacher* model to a strictly more efficient *student* model. KD is *lossless* if the transfer process yields a student that matches or outperforms its teacher across a broad spectrum of downstream tasks under an identical evaluation protocol.
>
> More formally, let $\mathcal{B} = \{b_1, \ldots, b_n\}$ be a set of benchmarks, $A_S(b)$ and $A_T(b)$ be the accuracies of the student and the teacher on benchmark $b$, respectively. A student is at least as good as the teacher up to a tolerance of $\alpha \in [0, 1]$:
>
> $$A_S(b) \geq (1 - \alpha)A_T(b)$$
>
> .
>
> $$\mathbf{1}_\alpha(b) = \begin{cases} 1 & \text{if } A_S(b) \geq (1 - \alpha)\, A_T(b), \\ 0 & \text{otherwise.} \end{cases} \tag{8}$$
>
> The *Win-and-Tie rate at a tolerance level of $\alpha$* of the student relative to the teacher is then
>
> $$C_\alpha = \frac{1}{|\mathcal{B}|} \sum_{b \in \mathcal{B}} \mathbf{1}_\alpha(b). \tag{9}$$
>
> We can now define *lossless distillation* when a student exhibits $C_0 \geq 0.5$, i.e. equal or better accuracy on at least half of the benchmarks without any tolerance. *$\alpha$-tolerant distillation* occurs if a student reaches a Win-and-Tie rate $C_\alpha \geq 0.5$ at tolerance level $\alpha$.
>
> The quality of a distillation process can be assessed by
>
> $$\alpha^\star = \inf\{\alpha \mid C_\alpha \geq 0.5\} \tag{10}$$
>
> Smaller $\alpha^\star$ indicates a more conservative and higher-quality distillation, as less tolerance is required for the student to match the teacher on at least half of the benchmarks.

**Win-and-Tie rate curves.** The definition above allows us to investigate the Win-and-Tie rate $C_\alpha$ as a function of of different values for $\alpha$ for each model. With increasing tolerance $\alpha$ the student is more often considered as equal or better than the teacher, which means increasing $C_\alpha$. If the student matches or outperforms the teacher on at least half of the benchmarks, the models can be considered at least equally performant and thus as a *successful distillation process*. Win-and-tie rate curves show at which tolerance values, successful distillation would be reached.

**Model comparison and Pareto front.** These curves also allow for model comparison: the higher the curve the better. The **Pareto-front** is at the top curve: at a given tolerance value, the best pick is the method that provides the best Win-and-Tie rate.

# B. Related Work

## B.1. Modern Recurrent and Hybrid Architectures.

**Linear attention and SSM alternatives to softmax attention.** A broad line of work targets sub-quadratic sequence operators with linear-time training and constant-memory decoding. Beyond state space models (SSMs) such as Mamba (Dao & Gu, 2024), this space includes data-dependent convolutions such as Hyena (Poli et al., 2023) and linear-attention families augmented with expressive gating mechanisms such as GLA, DeltaNet, Gated-DeltaNet, RWKV, and xLSTM (Yang et al., 2024; Schlag et al., 2021; Peng et al., 2023; Beck et al., 2024). Among recurrent gated variants, xLSTM offers two operators. *sLSTM* uses a scalar state with exponential input gating. *mLSTM* maintains a matrix-valued state with head-wise scalar gates and is fully parallelizable, with a recurrent formulation for decoding. These operators provide stable long-horizon

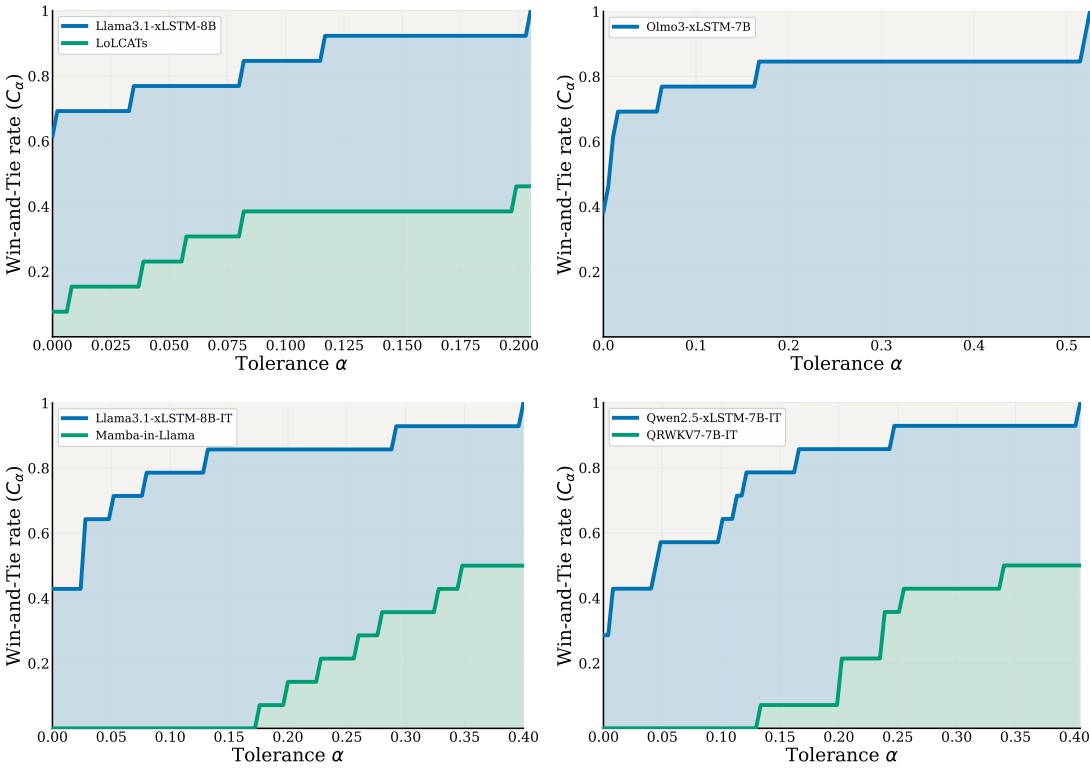

*Figure 6.* Win-and-Tie rate ($C_\alpha$) of our xLSTM-distilled Models (XLSTM-LLAMA3.1-8B, XLSTM-OLMO3-7B, XLSTM-LLAMA3.1-8B-IT, XLSTM-QWEN2.5-7B-IT) and their respective best performing subquadratic baselines across generation benchmarks spanning math, code, STEM, and chat domains (for OLMO3-7B there are no available Baselines at the time of writing). On the y-axis, we show the Win-and-Tie rate $C_\alpha$ between the student and teacher for a given tolerance $\alpha$. Distillation can be considered successful if the student matches or outperforms the teacher in 50% of the benchmarks. Thus, $C_\alpha$ values above 0.5 can be considered successful distillation attempts.

memory, efficient rank-one fast-weight updates, and constant-size states at inference.

**Hybrid models: inter-layer vs. intra-layer.** A growing literature blends quadratic attention with linear attention or SSM primitives. These works can be categorized into inter-layer and intra-layer hybrids.

*Inter-layer hybrids* interleave attention and linear sequence-mixing blocks. Notable early works include Jamba (Team, 2025b) and Zamba (Glorioso et al., 2024), which alternate between Mamba and global softmax attention layers. Samba (Ren et al., 2025) replaces the global attention layers in earlier designs with sliding-window attention, yielding a fully linear architecture that combines global memory with precise short-range recall. SWAX (Cabannes et al., 2025) alternates sliding-window attention and mLSTM and applies stochastic window resizing to strengthen long-context capability. Recent production-scale efforts extend this pattern. Nemotron-H (NVIDIA, 2025a) and the Nemotron-Nano-2 (NVIDIA, 2025b) replace most attention layers with Mamba. Qwen-Next (Team, 2025c) interleaves Gated DeltaNet layers and gated softmax attention layers. MiniMax-01 (MiniMax, 2025) combines Lightning Attention (Qin et al., 2024), a gated linear-attention variant, with global softmax attention layers. Recently, GPT-OSS combines global attention with sliding window attention (OpenAI, 2025).

*Intra-layer hybrids* fuse attention and linear or SSM branches within a block and mix outputs through addition or learned gates. Head-wise designs allocate some heads to attention and others to SSMs. Sequence-wise designs split tokens by absolute or relative position. Recent synchronous designs allow linear mixers and attention layers to process the full input sequence at the same time. Representative head-wise models include Hymba (Dong et al., 2025), which assigns half of the heads to parallel attention and the remainder to Mamba. Sequence-wise hybrids include TransMamba (Li et al., 2026), which can switch between attention and SSM mechanisms at different transition points for different layers. As an early influential SWA plus linear hybrid, BASED (Arora et al., 2024) utilized linear attention to compress tokens outside the sliding window into a compact global state.

Our method fits the intra-layer category. Every layer combines sliding-window attention with mLSTM. Unlike Hymba, we do not separate sliding-window and linear branches across heads. Each head models both local and global dependencies with both branches. We chose this design for two reasons. In distillation, the teacher's attention heads are already pre-trained, and assigning local or global roles would require analysis. The design also grants each head more expressive power. Unlike BASED, we do not restrict the linear branch to tokens that leave the sliding window. Both branches model the full input sequence and can exploit their complementarity. This design has recently also been shown to be effective for pre-training at smaller scales (Irie et al., 2025). We fuse the outputs by repurposing the mLSTM output gate, which yields a data-dependent combination of the two attention streams. In distillation, we also find that modeling sink tokens is important. Similar interventions appear in work on KV-cache compression (Xiao et al., 2024) and in hybrids such as Hymba. Unlike Hymba, which introduces learned meta tokens, we simply include the first four sink tokens in the sliding window. Recent work also seeks to mitigate the emergence of attention sinks through pre-training interventions (OpenAI, 2025; Miller, 2023). For those models, there is no need to handle sinks explicitly.

### B.2. Linearizing LLMs.

To reduce the prohibitive inference cost of Transformers on long sequences, recent work studies *linearization*, a post-training procedure that replaces some or all softmax self-attention layers in a pre-trained model with linear-time sequence mixers such as gated RNNs or state-space models. Notable examples include T2R (Kasai et al., 2021), SUPRA (Mercat et al., 2024), LoLCATs (Zhang et al., 2025), Mamba-in-Llama (Wang et al., 2024), RADLADS (Goldstein et al., 2025), MOHAWK (Bick et al., 2024) and Llamba (Bick et al., 2025). Most methods copy compatible weights from the teacher into the student, which yields far greater data efficiency than training from scratch. During the initial alignment stage, LoLCATs, MOHAWK, and Llamba match hidden states with an MSE loss, and MOHAWK and Llamba also match attention maps, which improves alignment but requires materializing the attention matrix with $O(n^2)$ memory. We follow the recipe of copying weights and then calibrating a small set of new gating and head-projection parameters with MSE.

Adaptation strategies divide into low-rank updates and full fine-tuning. LoLCATs and Lizard utilize low-rank adapters to reduce training cost. Mamba-in-Llama, Llamba, MOHAWK, and RADLADS perform full-model updates, which are more compute-intensive but reduce approximation error between student and teacher. Supervision choices also differ. LoLCATs and Lizard optimize next-token cross-entropy, while Mamba-in-Llama, Llamba, MOHAWK, and RADLADS add logit alignment with a KL loss term. We adopt a mixed objective that combines cross-entropy loss with a KL penalty, and we align only $k$ sampled logits so that teacher outputs can be precomputed once, similar to (Team, 2025a).

Architecturally, some systems hybridize linear sequence operators with variants of softmax attention, and others are fully softmax-free. LoLCATs and Lizard propose intra-layer hybrids that mix sliding-window attention with a linear path, with Lizard adding gated linear attention and global meta tokens. Mamba-in-Llama and MOHAWK study inter-layer hybrids in which only some layers are linearized (Bick et al., 2024). RADLADS and Llamba convert all attention to linear-time mixers. We follow the intra-layer route, pairing sliding-window attention with an mLSTM path and adding sink tokens, then gating the two paths for data-dependent mixing.

Training budgets and datasets vary widely, from about 20M to 12B tokens. RADLADS relies on DCLM (Li et al., 2024) and adds OpenThoughts (Guha et al., 2025) for hybrid reasoning models, MOHAWK uses C4 (Raffel et al., 2020), and Llamba reports gains from FineWeb-Edu (Penedo et al., 2024). Lizard and LoLCATs distill on small instruction datasets, for example, Alpaca. Our schedule uses about 650M tokens for hidden-state matching and 1.3B tokens for end-to-end fine-tuning on a Dolmino-derived mid-training mixture.

### B.3. Decentralized model training and weight merging.

A growing set of post-training pipelines decentralize (often within a research organization) by training multiple capability- or domain-specialized variants and then consolidating them into a single deployable model. Early on, Branch-Train-Merge (BTM) showed that independently trained domain experts can be ensembled or collapsed back into a single model via parameter averaging (Li et al., 2022). Recent large-scale systems follow related patterns: DeepSeek-V3.2 unifies expert behaviors by sampling from specialists and then performing SFT on the resulting traces to consolidate capabilities (DeepSeek-AI, 2025), while Command-A reports merging specialists in a way closely aligned with our proposed workflow (Cohere, 2025). Beyond direct weight fusion, MiMo Flash V2 uses multi-teacher on-policy distillation as a mechanism to combine models through teacher feedback on self-generated trajectories (Team, 2026).

When consolidation is performed in weight space, simple averaging can already be surprisingly effective (e.g., weight soups) (Wortsman et al., 2022), but more robust methods such as TIES explicitly mitigate parameter interference by resolving sign conflicts and trimming small-magnitude updates (Yadav et al., 2023).

## C. Extended Background

### C.1. Receptive Field of SWA

Although a depth-$L$ stack of SWA layers has a nominal receptive field of $LW$, in practice the effective receptive field grows much more slowly and is biased toward recent tokens. Empirical measurements, as well as signal-propagation arguments, suggest sublinear growth in $L$ (Xiao, 2025). This mirrors classic results for deep convolutional networks, where the effective receptive field is Gaussian-like and occupies only a small fraction of the theoretical context (Luo et al., 2016).

### C.2. Attention Sinks

Transformers often place large, persistent attention mass on initial tokens (e.g., `<BOS>`). Barbero et al. (2025) argue that sinks are a useful stabilizer that prevents over-mixing through depth and preserves token identity, explaining why *sink patterns* emerge broadly even when those tokens are semantically irrelevant. Sink behavior has been previously exploited for effective KV cache compression. StreamingLLM (Xiao et al., 2024) preserves a small sink prefix ($1-4$ tokens) plus a sliding window for recent context and evicts the remaining tokens in the cache. This compression partially recovers full-attention quality, improves length-generalization, and yields substantial decoding speedups. A complementary perspective is that row-wise softmax in attention forces every head to allocate its entire probability mass across the sequence, encouraging spurious focus on sinks. Minimal fixes have been proposed, replacing $\text{softmax}(x)_i = \frac{e^{x_i}}{\sum_j e^{x_j}}$ with $\text{softmax}(x)_i = \frac{e^{x_i}}{e^b + \sum_j e^{x_j}}$, where $b$ is either set to 0 or a learned bias (Miller, 2023; OpenAI, 2025). This is effectively equivalent to adding a no-op *null* key and value.

## D. Experimental & Implementation Details

All experiments were run on 8 H100 GPUs using PyTorch fully sharded data parallel (FSDP). We configured our training with a global batch size of 64 (using gradient accumulation), mixed precision (bfloat16 for operations, float32 for gradient all-gather), and gradient clipping at a threshold of 1.0 for full finetuning. To maximize GPU utilization, input sequences were packed to fill the maximum context length. We found that preserving the attention mask across these packed sequences,

*Table 1.* Hyperparams.

| Setting | Value |
|---|---|
| mLSTM/sLSTM configuration | `[1:0]` |
| mLSTM head dimension | 128 |
| Number of mLSTM heads | 32 |
| Position embeddings | true |
| Context size | 4096 |
| Total token budget | 20B (5B per expert) |
| Weight decay | 0.1 |
| Optimizer | AdamW |
| Batch size | 64 |
| Sequence packing | true |
| Learning rate (Phase I: MSE matching) | Cosine schedule, max LR 1e$-$2 |
| Learning rate (Phase II: Knowledge Distillation) | Warmup + LR (1e$-$5 Llama, 7e$-$7 Qwen) |
| CE loss weight | 0.9 |
| KL loss weight | 0.1 |
| Linear merge weights (Llama) | `math` $\cdot\, 0.35 +$ `code` $\cdot\, 0.35 +$ `stem` $\cdot\, 0.20 +$ `chat` $\cdot\, 0.10$ |
| Linear merge weights (Qwen) | `math` $\cdot\, 0.20 +$ `code` $\cdot\, 0.40 +$ `stem` $\cdot\, 0.15 +$ `chat` $\cdot\, 0.25$ |

rather than truncating it, improved performance for our hybrid architecture. This finding is consistent with prior work (Buitrago & Gu, 2025).

### D.1. Hyperparameters and Data Mixes

**Hyperparameters.** Table 1 summarizes the hyperparameters used throughout our distillation pipeline. We instantiate the student with an xLSTM [1:0] configuration (mLSTM-only), using 32 mLSTM heads with head dimension 128 and rotary position embeddings, and train at a context length of 4096. Optimization is performed with AdamW (weight decay 0.1) at a global batch size of 64. We enable sequence packing to maximize context utilization while preserving attention masks across packed sequences, which improves throughput without truncating examples. In **Phase I**, we perform layer-wise hidden-state alignment with an MSE objective (cosine learning-rate schedule with peak LR $10^{-2}$), keeping the optimization focused on the newly introduced mixer/gating parameters. In **Phase II**, we switch to end-to-end knowledge distillation with a warmup followed by a low constant learning rate ($10^{-5}$ for LLAMA, $7 \times 10^{-7}$ for QWEN). The Phase II stage interpolates between next-token cross-entropy and KL distillation, using weights $\gamma = 0.9$ (*CE*) and $\beta = 0.1$ (*KL*).

**Data Mix.** Table 2 outlines the domain-specific data mixtures used to train the individual experts (*math*, *code*, *stem*, *chat/instruction-following*), as well as the multi-task mixture used for a *generalist*-XLSTM student. Across both LLAMA and QWEN variants, the data mixes are mostly constructed from *Nemotron Nano Pre- and Post-training* datasets. Moreover, for some of the experts (e.g., code), we enhance the datamix with appealing domain-targeted data, such as *OpenCodeInstruct* and *Dolci-Think-SFT*. The stem expert differs the most between the two model families; the LLAMA stem expert was trained solely on the *Nemotron STEM Post-Training* dataset, whereas for QWEN we enhanced our datamix with more specialized *STEM/MCQ* traces, to minimize the gap to the teacher.

### D.2. Generalist vs. Merged Expert Student

To assess the effectiveness of decentralized expert training, we compare a *generalist* student trained on the multi-task mixture against a *merged expert* student obtained by training four domain specialists and combining them via linear merging. In the expert setting, each specialist is trained on its corresponding domain mixture (Table 2) for ~5B tokens, for a total of 20B tokens, and then merged using the fixed weights in Table 1. In the generalist setting, a single student is trained for the same 20B token budget on the multi-task mixture, keeping architecture and optimization matched.

Table 7 shows that the *generalist* student trained on the multi-task mixture lags behind the *merged-expert* student, with the largest gaps appearing on specialized evaluations that directly probe individual capabilities. XLSTM-LLAMA3.1-8B-IT consistently outperforms XLSTM-LLAMA3.1-8B-IT-GENERALIST on math and reasoning (MATH500 0.54 vs. 0.37, MATH 0.55 vs. 0.37), code generation (HumanEval 0.63 vs. 0.50, HumanEval+ 0.56 vs. 0.47), and instruction-following

*Table 2.* Data mixes for expert and multi-task linearization.

| Model | Dataset | Split | Mixing Weight (%) |
|---|---|---|---|
| *xLSTM-Llama Math Expert* | | | |
| | nvidia/Nemotron-Pretraining-SFT-v1 | math | 50 |
| | nvidia/Llama-Nemotron-Post-Training-Dataset | math | 25 |
| | nvidia/Nemotron-Post-Training-Dataset-v2 | math | 25 |
| *xLSTM-Llama Code Expert* | | | |
| | nvidia/Llama-Nemotron-Post-Training-Dataset | code | 21 |
| | nvidia/Nemotron-Post-Training-Dataset-v2 | code | 1 |
| | nvidia/Nemotron-Pretraining-SFT-v1 | code | 64 |
| | nvidia/OpenCodeInstruct | | 11 |
| | allenai/Dolci-Think-SFT-Python | | 3 |
| *xLSTM-Llama STEM Expert* | | | |
| | nvidia/Nemotron-Post-Training-Dataset-v1 | stem | 100 |
| *xLSTM-Llama STEM Expert FT* | | | |
| | nvidia/Nemotron-Post-Training-Dataset-v2 | stem | 100 |
| *xLSTM-Llama Instruction Following Expert* | | | |
| | nvidia/Nemotron-Pretraining-SFT-v1 | general | 90 |
| | nvidia/Nemotron-Post-Training-Dataset-v1 | chat | 10 |
| *xLSTM-Llama Instruction Following Expert FT* | | | |
| | nvidia/Nemotron-Post-Training-Dataset-v2 | chat | 100 |
| *xLSTM-Qwen Math Expert* | | | |
| | nvidia/Nemotron-Pretraining-SFT-v1 | math | 76 |
| | nvidia/Llama-Nemotron-Post-Training-Dataset | math | 20 |
| | nvidia/Nemotron-Post-Training-Dataset-v2 | math | 4 |
| *xLSTM-Qwen Code Expert* | | | |
| | nvidia/Llama-Nemotron-Post-Training-Dataset | code | 20 |
| | nvidia/Nemotron-Post-Training-Dataset-v2 | code | 1.2 |
| | nvidia/Nemotron-Pretraining-SFT-v1 | code | 60 |
| | nvidia/OpenCodeInstruct | | 12 |
| | allenai/Dolci-Think-SFT-Python | | 6.8 |
| *xLSTM-Qwen STEM Expert* | | | |
| | nvidia/Nemotron-Pretraining-Specialized-v1 | | 90.6 |
| | nvidia/Nemotron-Science-v1 | MCQ | 3.4 |
| | nvidia/Nemotron-Post-Training-Dataset-v2 | stem | 6 |
| *xLSTM-Qwen Instruction Following Expert* | | | |
| | nvidia/Nemotron-Pretraining-SFT-v1 | general | 88 |
| | nvidia/Nemotron-Post-Training-Dataset-v2 | chat | 8 |
| | nvidia/Nemotron-Instruction-Following-Chat-v1 | chat_if | 4 |
| *Multi-task* | | | |
| | nvidia/Nemotron-Pretraining-SFT-v1 | math | 19 |
| | nvidia/Llama-Nemotron-Post-Training-Dataset | math | 5 |
| | nvidia/Nemotron-Post-Training-Dataset-v2 | math | 1 |
| | nvidia/Nemotron-Pretraining-Specialized-v1 | stem | 22.66 |
| | nvidia/Nemotron-Science-v1 | MCQ | 0.84 |
| | nvidia/Nemotron-Post-Training-Dataset-v2 | stem | 1.5 |
| | nvidia/Nemotron-Pretraining-SFT-v1 | general | 22 |
| | nvidia/Nemotron-Post-Training-Dataset-v2 | chat | 2 |
| | nvidia/Nemotron-Instruction-Following-Chat-v1 | chat_if | 1 |
| | nvidia/Llama-Nemotron-Post-Training-Dataset | code | 5 |
| | nvidia/Nemotron-Post-Training-Dataset-v2 | code | 0.3 |
| | nvidia/Nemotron-Pretraining-SFT-v1 | code | 15 |
| | nvidia/OpenCodeInstruct | | 3 |
| | allenai/Dolci-Think-SFT-Python | | 1.7 |

(IFEval 0.69 vs. 0.49, MT-Bench 6.05 vs. 5.45). xLSTM-QWEN2.5-7B-IT similarly remains ahead of xLSTM-QWEN2.5-7B-IT-GENERALIST across the same specialized benchmarks (e.g., MATH 0.66 vs. 0.62, MATH Level 5 0.42 vs. 0.36, GPQA-D 0.26 vs. 0.19, MT-Bench 5.96 vs. 5.55).

We attribute this behavior to the multi-task distillation. Although the total token budget for distillation is the same, the generalist allocates fewer updates to the specialized traces that matter for our benchmarks (math, code, and instruction-format data), whereas each expert is trained on a single-domain data distribution. Linear merging then retains much of this specialization in the merged checkpoint, in line with decentralized post-training recipes (Cohere, 2025).

### D.3. Merge-Weight Selection Protocol

Merge weights $\lambda_i$ in Eq. (7) are selected without any access to the reported test sets. We use three sources of signal, applied in order of preference per benchmark.

**(i) Held-out validation splits.** For benchmarks that ship with a validation split, we tune merge weights on that split only. This applies to GSM8K (`train`/`validation` split), MATH (the official `train` split, from which we hold out a fixed subset), and MBPP (the official `validation` split). Test splits are used exclusively for the numbers reported in Figure 4 and the appendix tables.

**(ii) Held-out distillation data.** For domains without a validation split, we tune on held-out subsets of the domain-specific distillation mixtures (Table 2) using validation cross-entropy and perplexity as the selection signal. This isolates merge-weight selection from the evaluation suite.

**(iii) Heuristics.** On top of (i)–(ii), we apply two simple heuristics: (a) experts that underperform across multiple domains are downweighted, and (b) we prefer near-uniform weights as a regularizer against overfitting on small validation sets.

**Benchmarks not used for tuning.** The following test-only benchmarks were *not* used at any point during merge-weight selection: HumanEval, HumanEval+, CruxEval, GPQA, GPQA-Diamond, IFEval, MT-Bench, MMLU, and the language-understanding suite reported in Table 5. The fact that the resulting merged checkpoints recover competitive performance on these unseen benchmarks is consistent with merge weights that generalize across domains rather than overfit to the evaluation suite.

## E. Additional Results

In this section, we provide additional details on experiments and provide additional results to complement the main text.

### E.1. Sink Analysis

As described in Section 2, Transformers often place large, persistent attention mass on the first token or on a small set of initial tokens. Barbero et al. (2025) argue that sinks are a useful stabilizer that prevents over-mixing through depth and preserves token identity. Consequently, many established LLMs exhibit sink patterns if they are not equipped with counter-measurements to prevent them from emerging. To illustrate the sink patterns in our Llama 3.1 8B teacher model, in Figure 7 we plot attention maps for two layers and two heads in a 3-dimensional grid. Indeed, we find that the Transformer teacher puts the majority of its attention mass on the first token. In contrast, when analyzing the distilled mLSTM-only checkpoints, we find that they struggle to represent the sink patterns present in the pre-trained teacher model. We observe that this forgetting effect worsens as the input sequence length grows. We suspect this is attributable to the decaying effect of the mLSTM's forget gate.

Therefore, modeling sink behavior is critical for strong performance. In our experiments, we empirically confirmed that modeling sink patterns in combination with the sliding window results in a considerably lower loss and stronger downstream performance, as illustrated in our components ablation in Figure 15.

### E.2. Output Gate Analysis

As described in Section 3 we combine the individual outputs of the mLSTM and SWA components via a data-dependent output gate. To better understand the relative contributions of each component to the final prediction of our student models, we analyze the activations of the output gate. To this end, we forward 128 sequences (sequence length 4096) through our model and record the average activation values for each layer and attention head, resulting in a layer $\times$ head gating matrix.

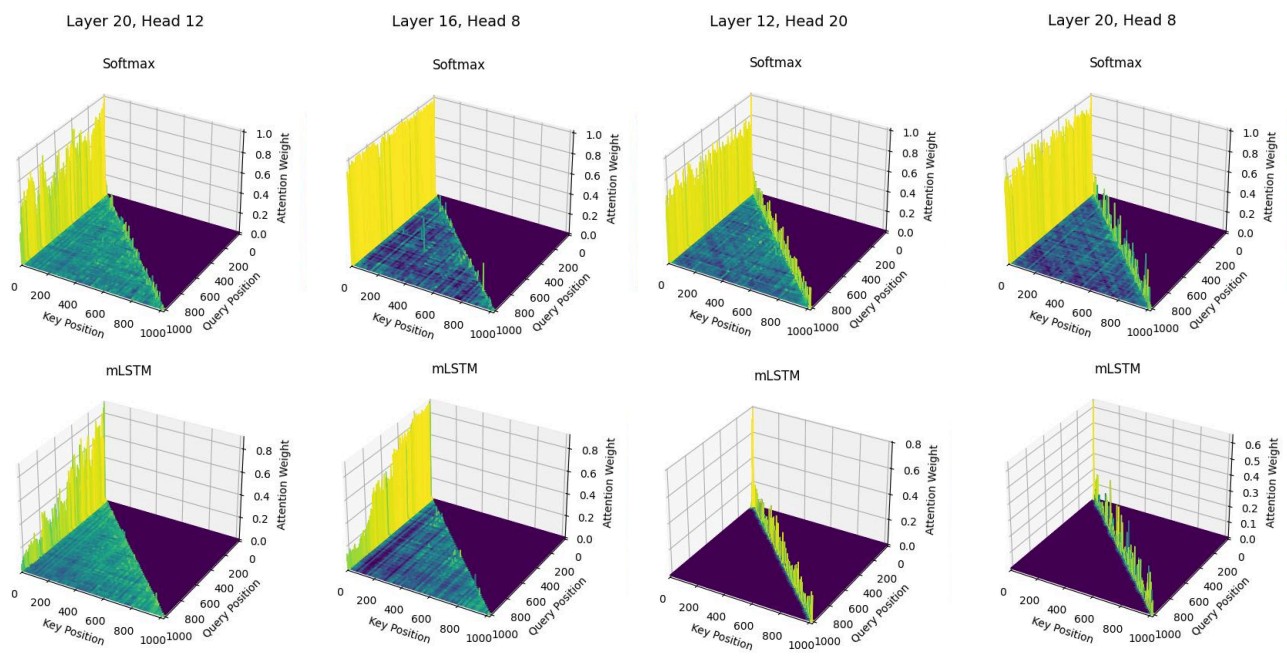

*Figure 7.* Illustration of **attention sinks** in the Llama 3.1 8B teacher model.

In Figure 8, we observe that both components contribute significantly to the final output prediction across all layers. Noticeably, mLSTM dominates in the first two layers, suggesting that the global contextual information carried by mLSTM blocks is integrated early on in the layer stack. The middle layers (3 – 16) are predominantly influenced by SWA, while the final layers (17 – 32) exhibit a more balanced contribution from both components, with neither clearly dominating.

### E.3. Downstream Evals

After training, we evaluate our distilled student models on downstream tasks, which we group into 4 categories:

- **Language Understanding**: log-likelihood and commonly used multiple-choice benchmarks

- **Language Generation & Reasoning**: mathematics, coding and other established reasoning benchmarks

- **Language Generation Quality via MT-Bench**: across 6 tasks contained in MT-Bench (Zheng et al., 2023)

- **Needle in a Haystack**: we evaluate long context retrieval via Needle in a Haystack tasks.

We conduct all our evaluations using `lm-eval` released by Gao et al. (2024). To ensure a consistent and fair comparison, we maintain the same number of few-shot examples, context lengths, and generation budgets across all teacher and student models. Where available, we adopt the same evaluation settings as used by Lambert et al. (2024) and Touvron et al. (2023). We provide all evaluation configurations for `base` and `instruct` models in Tables 3 and 4, respectively.

**Language Understanding & Knowledge.** To complement the relative teacher scores reported in Figure 3, we report the raw performance scores across all language understanding tasks for all student and teacher models in Table 5. We find that our distilled student models match and in some tasks even slightly exceed their respective teacher performances. In contrast, other linearization recipes fall short of their respective teachers, exhibiting a significant performance gap.

**Language Generation & Reasoning.** Similarly, we report the raw performance scores across all language generation and reasoning tasks for all student and teacher models in Table 6. Again, we observe that our distilled students almost match their respective teacher performances, while alternative distillation recipes fall short.

**MT-bench.** Figure 9 shows MT-Bench (Zheng et al., 2023) performance as evaluated by GPT-5.1 broken down by category.

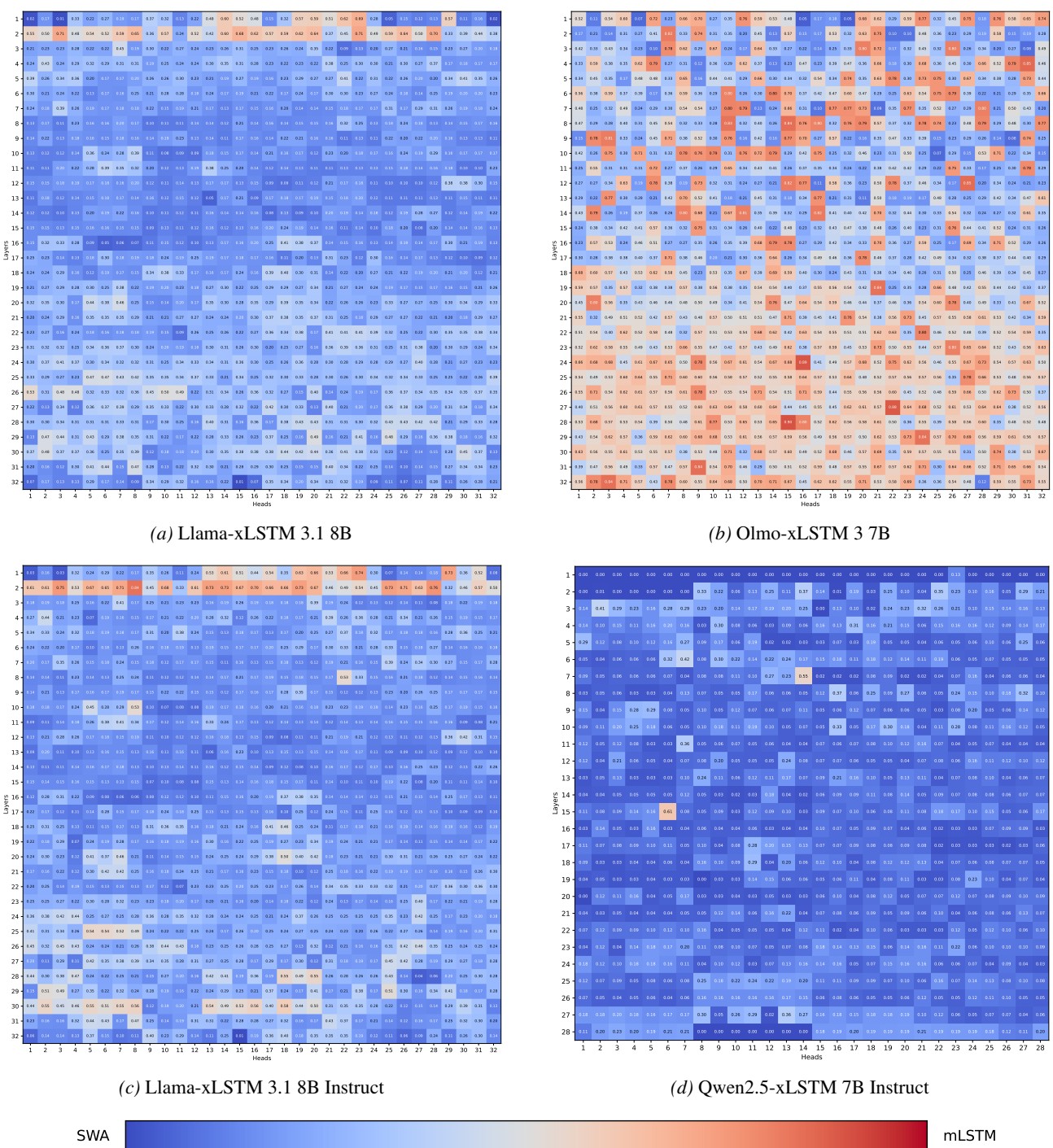

*(a)* Llama-xLSTM 3.1 8B

*(b)* Olmo-xLSTM 3 7B

*(c)* Llama-xLSTM 3.1 8B Instruct

*(d)* Qwen2.5-xLSTM 7B Instruct

*Figure 8.* Illustration of the data-dependent **output gate** output matrix. After training, we analyze how the mLSTM and SWA contributions are mixed across layers and heads. The plots above show the median output gate's sigmoid activation across layers and heads over 128 randomly drawn data samples with a context size of 4096. For a given example, 0 (blue) indicates that only sliding window is used, whereas an activation of 1 (red) means only mLSTM is used. We observe balanced to high mLSTM activations across model families.

*Table 3.* Downstream **evaluation configurations** for language understanding, generation, and quality benchmarks for all student and teacher `base` models.

| Task | # of shots | Generation Budget |
|------|-----------|-------------------|
| *Language Understanding* | | |
| PIQA | 0 | – |
| ARC-e | 0 | – |
| ARC-c | 25 | 100 |
| HellaSwag | 10 | – |
| Winogrande | 5 | – |
| MMLU | 5 | 10 |
| *Language Generation* | | |
| GPQA Diamond CoT | 0 | 2048 |
| GPQA Main CoT | 0 | 2048 |
| GSM8K | 8 | 1024 |
| HumanEval | 0 | 1024 |
| HumanEval Plus | 0 | 1024 |
| MBPP | 3 | 256 |
| MBPP Plus | 3 | 1024 |
| *Language Quality* | | |
| MT-Bench | – | 1024 |

*Table 4.* Downstream **evaluation configurations** for language understanding, generation, and quality benchmarks for all student and teacher `instruct` models.

| Task | # of shots | Gen Budget | Chat | Multiturn ICL |
|------|-----------|-----------|------|---------------|
| *Language Understanding* | | | | |
| PIQA | 0 | – | ✓ | ✗ |
| ARC-e | 0 | – | ✓ | ✗ |
| ARC-c | 25 | 100 | ✓ | ✓ |
| HellaSwag | 10 | – | ✓ | ✓ |
| Winogrande | 5 | – | ✓ | ✓ |
| MMLU | 5 | 10 | ✓ | ✓ |
| *Language Generation* | | | | |
| GPQA Diamond CoT | 0 | 2048 | ✓ | ✗ |
| GPQA Main CoT | 0 | 2048 | ✓ | ✗ |
| GSM8K CoT | 5 | 3072 | ✓ | ✓ |
| MATH 500 | 0 | 3072 | ✓ | ✓ |
| MATH | 0 | 3072 | ✓ | ✓ |
| Math Level 5 | 4 | 1024 | ✓ | ✓ |
| HumanEval (64) Instruct | 0 | 3072 | ✓ | ✗ |
| HumanEval+ (64) Instruct | 0 | 3072 | ✓ | ✗ |
| MBPP | 3 | 1024 | ✓ | ✓ |
| MBPP+ | 3 | 1024 | ✓ | ✓ |
| Cruxeval-O | 0 | 3072 | ✓ | ✓ |
| Cruxeval-I | 2 | 3072 | ✓ | ✓ |
| *Language Quality* | | | | |
| MT-Bench | – | 1024 | ✓ | ✓ |

*Table 5.* Raw scores for **downstream evaluations** on **language understanding** tasks. Our models perform comparably to their LLAMA3.1-8B, OLMO3-7B, LLAMA3.1-8B-IT, and QWEN2.5-7B-IT teacher models across language understanding tasks.

| Model | PIQA↑ | ARC-e↑ | ARC-c↑ | HellaSwag↑ | Winogrande↑ | MMLU↑ | Avg.↑ |
|---|---|---|---|---|---|---|---|
| **T**: QWEN2.5-7B(Yu et al., 2025) | 78.7 | 80.3 | **63.7** | **80.2** | **76.4** | **75.3** | **75.8** |
| **S**: QRWKV6-7B(Goldstein et al., 2025) | **79.4** | **80.6** | 60.2 | 77.9 | 73.9 | 65.4 | 72.9 |
| **T**: LLAMA3.1-8B(Team, 2024) | **80.1** | 81.6 | **57.8** | 81.9 | 76.9 | 65.9 | 74.0 |
| **S**: LoLCATs(Zhang et al., 2025) | 79.5 | **83.2** | 54.6 | 75.3 | 74.0 | 52.9 | 69.9 |
| **S**: xLSTM-LLAMA3.1-8B | 80.0 | 82.9 | 55.9 | **82.3** | **79.0** | **66.1** | **74.4** |
| **T**: OLMO3-7B(Olmo, 2025) | **78.1** | 80.7 | **58.4** | 56.7 | **73.4** | **66.4** | **69.0** |
| **S**: xLSTM-OLMO3-7B | 77.9 | **81.1** | 57.8 | **56.9** | 72.8 | 65.7 | 68.7 |
| **T**: QWEN2.5-7B-IT(Yu et al., 2025) | 74.5 | 68.6 | 59.4 | 74.5 | 64.2 | **74.5** | 69.3 |
| **S**: QRWKV7-7B-IT(Goldstein et al., 2025) | 76.9 | 74.9 | **62.0** | **78.1** | 69.9 | 68.2 | **71.7** |
| **S**: xLSTM-QWEN2.5-7B-IT | **79.4** | **80.2** | 60.8 | 60.0 | 74.7 | 73.7 | 71.4 |
| **S**: xLSTM-QWEN2.5-7B-IT-GENERALIST | 79.1 | 73.7 | 54.1 | 58.9 | **75.4** | 66.6 | 68.7 |
| **T**: LLAMA3-8B-IT(Team, 2024) | 77.3 | **76.9** | **66.2** | 72.4 | **73.9** | **68.6** | **72.6** |
| **S**: MAMBA-IN-LLAMA(Wang et al., 2024) | **82.8** | 74.8 | 62.1 | 64.1 | 59.2 | 56.8 | 66.6 |
| **T**: LLAMA3.1-8B-IT(Team, 2024) | **79.9** | 81.8 | 56.7 | **59.3** | 78.0 | 68.9 | 70.1 |
| **S**: xLSTM-LLAMA3.1-8B-IT | 79.8 | 81.2 | 56.0 | 58.7 | 76.9 | 68.0 | **71.4** |
| **S**: xLSTM-LLAMA3.1-8B-IT-GENERALIST | 78.8 | 71.1 | **58.4** | 57.9 | 73.2 | **72.8** | 67.9 |

*Table 6.* Raw scores for **downstream evaluations** on **language generation tasks**. Our models perform comparably or slightly exceed their respective LLAMA3.1-8B and OLMO3-7B teacher models, while significantly outperforming distilled models with alternative linearization recipes.

| Model | GPQA-D (0)↑ | GPQA (0)↑ | GSM8K (8)↑ | HumanEval (0)↑ | HumanEval+ (0)↑ | MBPP (3)↑ | MBPP+ (3)↑ | Avg.↑ |
|---|---|---|---|---|---|---|---|---|
| **T**: QWEN2.5-7B(Yu et al., 2025) | **32.3** | **30.8** | **80.8** | **65.9** | **47.6** | **62.6** | **80.4** | **57.2** |
| **S**: QRWKV6-7B(Goldstein et al., 2025) | 9.60 | 13.2 | 23.8 | 20.7 | 33.5 | 0.0 | 27.0 | 14.4 |
| **T**: LLAMA3.1-8B(Team, 2024) | 10.6 | 13.8 | 48.4 | 35.4 | **29.9** | 48.4 | 61.9 | 35.5 |
| **S**: LoLCATs(Zhang et al., 2025) | 1.77 | 2.23 | 3.87 | 2.13 | 3.35 | 0.0 | 0.0 | 1.9 |
| **S**: xLSTM-LLAMA3.1-8B | **17.7** | **15.8** | **57.8** | **39.0** | 23.8 | 42.8 | 56.9 | **36.3** |
| **T**: OLMO3-7B(Olmo, 2025) | **20.7** | 19.0 | **67.5** | **32.9** | **28.7** | **50.6** | **71.7** | **41.6** |
| **S**: xLSTM-OLMO3-7B | 17.7 | **19.4** | 56.6 | 32.3 | 27.4 | 36.8 | 52.4 | 34.7 |

*Table 7.* Raw scores for **downstream evaluations** on **language generation tasks**. Our models perform comparably to their respective LLAMA3.1-8B-IT and QWEN2.5-7B-IT teacher models, while significantly outperforming distilled models with alternative linearization recipes.

| Model | GSM8K (8)↑ | MATH500 (0)↑ | MATH (0)↑ | MATH Level 5 (0)↑ | HumanEval (0)↑ | HumanEval+ (0)↑ | MBPP (3)↑ | MBPP+ (3)↑ | CruxEval-O (0)↑ | CruxEval-I (0)↑ | GPQA-D (0)↑ | GPQA (0)↑ | IfEval (0)↑ | MT-Bench (0)↑ | Avg.↑ |
|---|---|---|---|---|---|---|---|---|---|---|---|---|---|---|---|
| **T**: QWEN2.5-7B-IT(Yu et al., 2025) | **0.90** | **0.74** | **0.74** | **0.49** | **0.81** | **0.74** | 0.61 | **0.79** | 0.41 | 0.41 | **0.34** | **0.36** | **0.74** | 5.20 | 0.95 |
| **S**: QRWKV7-7B-IT(Goldstein et al., 2025) | 0.53 | 0.40 | 0.39 | 0.12 | 0.62 | 0.55 | 0.49 | 0.69 | 0.18 | 0.22 | 0.22 | 0.19 | 0.59 | 3.96 | 0.65 |
| **S**: xLSTM-QWEN2.5-7B-IT | **0.90** | 0.65 | 0.66 | 0.42 | 0.78 | 0.71 | **0.63** | **0.79** | **0.42** | **0.42** | 0.26 | 0.22 | 0.67 | **5.96** | **0.96** |
| **S**: xLSTM-QWEN2.5-7B-IT-GENERALIST | 0.89 | 0.63 | 0.62 | 0.36 | 0.76 | 0.68 | 0.62 | 0.78 | 0.37 | 0.36 | 0.19 | 0.14 | 0.62 | 5.55 | 0.90 |
| **T**: LLAMA3.1-8B-IT(Team, 2024) | **0.85** | 0.51 | 0.50 | 0.19 | **0.63** | **0.57** | **0.59** | 0.69 | 0.26 | **0.28** | **0.30** | **0.32** | **0.78** | 5.08 | 0.83 |
| **S**: xLSTM-LLAMA3.1-8B-IT | 0.83 | **0.54** | **0.55** | 0.22 | **0.63** | 0.56 | 0.54 | 0.66 | **0.34** | 0.27 | 0.21 | 0.20 | 0.69 | **6.05** | **0.88** |
| **S**: xLSTM-LLAMA3.1-8B-IT-GENERALIST | 0.82 | 0.37 | 0.37 | **0.22** | 0.50 | 0.47 | 0.54 | **0.71** | 0.29 | 0.25 | 0.16 | 0.14 | 0.49 | 5.45 | 0.77 |
| **T**: LLAMA3-8B-IT(Team, 2024) | **0.82** | **0.29** | **0.29** | **0.09** | **0.57** | **0.54** | **0.56** | **0.74** | **0.26** | **0.20** | **0.34** | **0.29** | **0.77** | 5.50 | 0.80 |
| **S**: MAMBA-IN-LLAMA(Wang et al., 2024) | 0.68 | 0.12 | 0.12 | 0.04 | 0.38 | 0.34 | 0.33 | 0.41 | 0.09 | 0.16 | 0.25 | 0.22 | 0.52 | 3.97 | 0.54 |

\* 50% attention layers

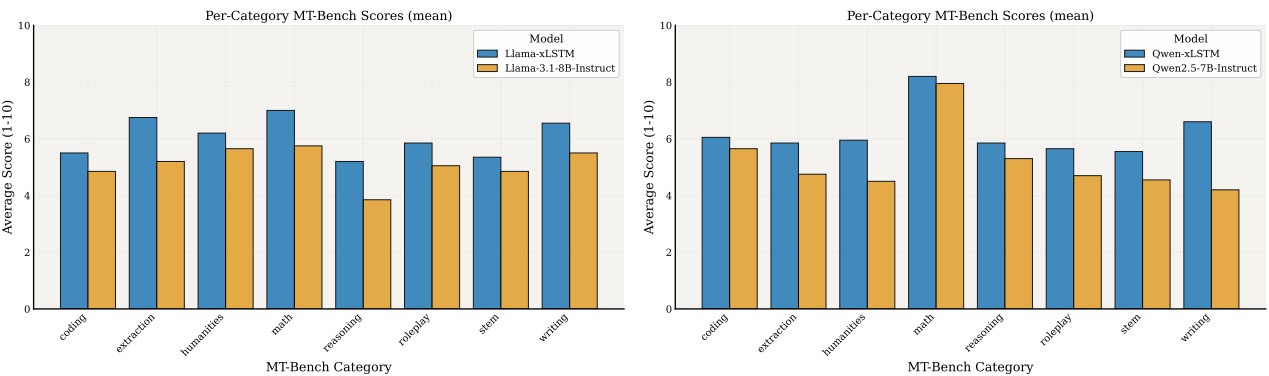

*Figure 9.* MT-Bench performance per category as judged by GPT-5.1.

*Table 8.* Comparison of LLAMA3.1-8B-IT and QWEN2.5-7B-IT against our Students on Needle-in-a-Haystack tasks.

| Model | NIAH Single | | | | NIAH Multi | | | |
|---|---|---|---|---|---|---|---|---|
| | **1024** | **4096** | **8192** | **16384** | **1024** | **4096** | **8192** | **16384** |
| LLAMA3.1-8B-IT | 1.000 | 1.000 | 1.000 | 1.000 | 1.000 | 1.000 | 1.000 | 1.000 |
| xLSTM-LLAMA3.1-8B-IT | 0.686 | 0.148 | 0.078 | 0.024 | 0.664 | 0.130 | 0.088 | 0.034 |
| QWEN2.5-7B-IT | 0.998 | 0.998 | 1.000 | 1.000 | 1.000 | 1.000 | 1.000 | 0.998 |
| xLSTM-QWEN2.5-7B-IT | 0.656 | 0.140 | 0.078 | 0.024 | 0.640 | 0.116 | 0.088 | 0.034 |

Both xLSTM-QWEN2.5-7B-IT and xLSTM-LLAMA3.1-8B-IT outperform their teacher models on all 8 MT-bench categories.

**Needle in a Haystack.** In Appendix Table 8, we report Needle-in-a-Haystack (NIAH; (Hsieh et al., 2024)) accuracy for both the single-needle and multi-needle variants across four context lengths (1k, 4k, 8k, and 16k tokens). Overall, the instruction-tuned baselines (LLAMA3.1-8B-IT and QWEN2.5-7B-IT) maintain near-perfect recall across all lengths, whereas our distilled Students degrade with increasing context, with the largest drop occurring at 4k and compounding further at 8k and 16k.

### E.4. Inference Time Analysis

In this section, we provide additional details on our inference time tests.

**Setup.** We run all our inference time tests on a single H100 GPU with 80GB of memory. Our implementations for both student and teacher are based on the `transformers` library Wolf et al. (2020) and their respective classes for Llama 3 (Touvron et al., 2023). For our hybrid student model, we replace the self-attention mechanism of the teacher model with our hybrid mechanism of mLSTM and SWA. To accelerate runtimes, we leverage `torch.compile` using a static KV-cache. For the Transformer-based teacher, we leverage FlashAttention-2 (Dao, 2024). Similarly, for our hybrid student, we make use of the Triton kernels released by Beck et al. (2025a) for the mLSTM part and FlashAttention for the sliding window part. To enable compilation of our hybrid student via `torch.compile`, we utilize a custom static cache implementation that retains both the mLSTM states and the relevant keys/values of sink tokens and the sliding window.

**Prefill vs. Generation.** We separate our inference time tests into two stages: prefilling and generation. While the prefilling stage encodes the input prompt by the user and populates the KV cache, the generation or decoding stage autoregressively samples tokens until sequence termination, starting from the pre-filled KV cache (Pope et al., 2023). Our inference tests are characterized by three core hyperparameters, the batch size $B$, the context length $C$, and the generation budget $G$. Consequently, if one would only perform prefilling without generating any tokens, then $G = 0$. Similarly, if we only perform generation without any prefill sequence, then $C = 0$ These two scenarios are reflected in Figure 10a and Figure 11a, respectively. For every combination of $B$, $C$, and $G$, we always first conduct three warmup runs, which include the compilation of our model. Afterwards, we record runtimes, memory consumption, and throughput across five runs and average metrics across them.

**Generation Latency & Memory Consumption.** To complement the results that we presented in Section 4.4, we report additional metrics for generation latency and memory consumption across varying batch sizes $B \in [1, 4, 8]$ and sequence lengths $C \in [128, 1024, 4096, 16384, 32768, 65536, 131072]$ in Figures 12 and 13, respectively. The purpose of this experiment is to better understand how the inference time advantages behave with increasing batch sizes and sequence lengths.

In Figure 12, we make two important observations. First, our hybrid student only exhibits a slight increase in total inference latency when increasing the batch size from 1 to 8. This is because its computational complexity does not grow with the sequence length, due to the recurrent inference mode of mLSTM and the fixed sliding window of 256 tokens. Consequently, for the largest batch size we compare ($B = 8$), the computation remains memory-bound. Second, we observe that the computational demand of the Transformer-based teacher grows faster with increasing sequence length, due to the quadratic complexity of self-attention. For example, when increasing the batch size from 1 to 4, the runtime at $C = 65K$ of our teacher model increases two-fold. Similarly, the required memory grows quickly as the batch size increases and the KV cache gets larger, causing the model to out of memory (OOM), as indicated by missing dots for the teacher. The differences in RAM consumption are further highlighted in Figure 13. The RAM consumption of the Transformer-based teacher grows

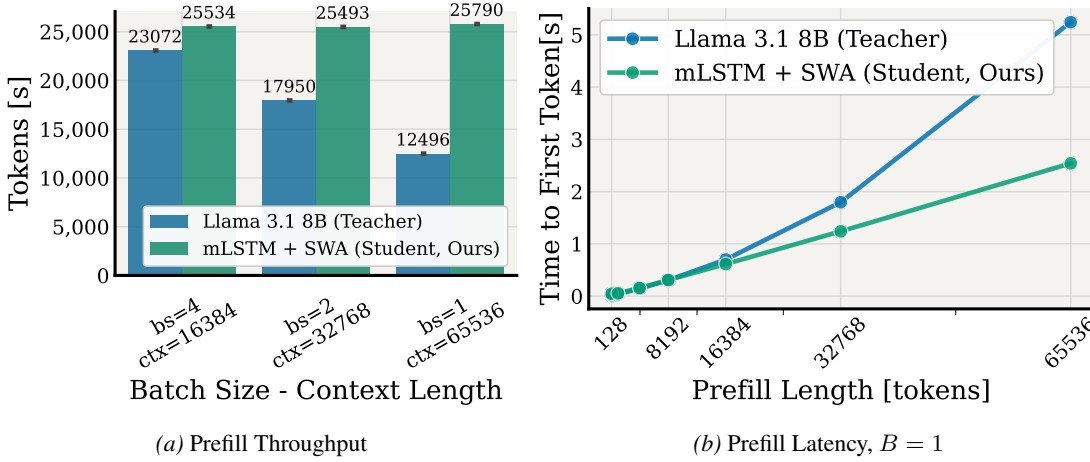

*(a)* Prefill Throughput

*(b)* Prefill Latency, $B = 1$

*Figure 10.* Inference comparison for the **prefilling stage** between the Transformer-based teacher and our mLSTM-based student. In **(a)**, we report prefill throughput for varying context lengths and batch sizes. In **(b)**, we show the prefilling latency for varying prefill lengths and $B = 1$.

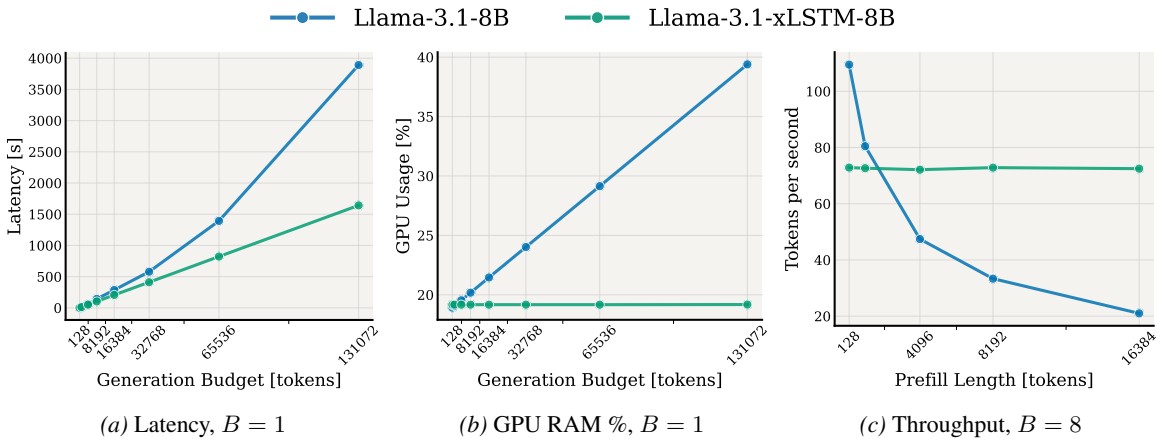

*(a)* Latency, $B = 1$

*(b)* GPU RAM %, $B = 1$

*(c)* Throughput, $B = 8$

*Figure 11.* Inference comparison for the **generation stage** between the Transformer-based teacher and our xLSTM-based student. In **(a)**, we show generation latency at different generation budgets ($B = 1$). In **(b)**, we report the memory consumption in % of GPU memory during the generation ($B = 1$). In **(c)**, we show the generation throughput when generating 100 tokens with varying prefill lengths and $B = 8$.

quickly with increasing sequence length and batch size. In contrast, our hybrid student remains constant along the sequence length and only exhibits a slight increase with larger batch sizes due to the constant memory complexity.

**Generation Throughput**. Finally, we report the average generation throughputs for a fixed generation budget of $G = 100$ tokens with varying prefill lengths and $B \in [1, 4, 8]$ in Figure 14. Again, we observe significantly higher throughputs for our hybrid student of up to almost $4\times$ that of the Transformer-based teacher as the sequence length increases. Note that due to OOMs, we only show metrics for $B = 4$ and $B = 8$ up to $C = 32K$ and $C = 16K$, respectively.

# F. Ablations

In this section, we empirically analyse a variety of important components of our distillation recipe.

## F.1. Effects of SWA, Attention Sinks & Gating

First, we ablate components of our hybrid attention operator under a fixed linearization recipe. We compare four variants: (i) pure linear attention, (ii) mLSTM, (iii) mLSTM + SWA, and (iv) mLSTM + SWA + sink tokens. For SWA, we use

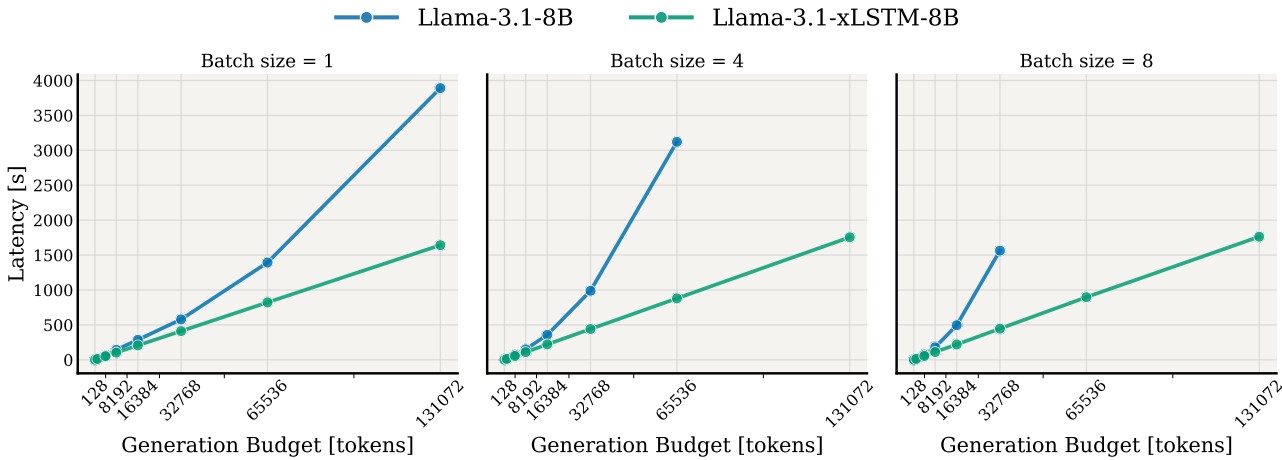

*Figure 12.* **Latency**. We report the latency for generation with varying token generation budgets and batch sizes. Our mLSTM-based student exhibits lower generation latency than the Transformer-based teacher. This advantage grows with larger generation budgets and batch sizes. Missing dots for the teacher indicate OOM.

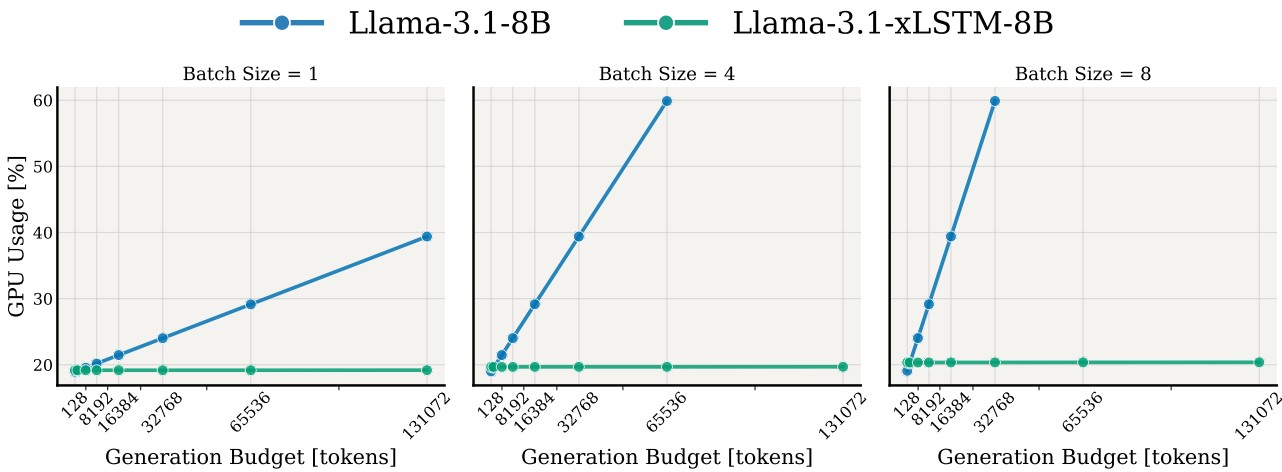

*Figure 13.* **GPU RAM**. We report the memory consumption in % of GPU memory during the generation for varying batch sizes. Our mLSTM-based student requires significantly less memory compared to the Transformer-based teacher. This advantage grows with larger generation budgets and batch sizes. Missing dots for the teacher indicate OOM.

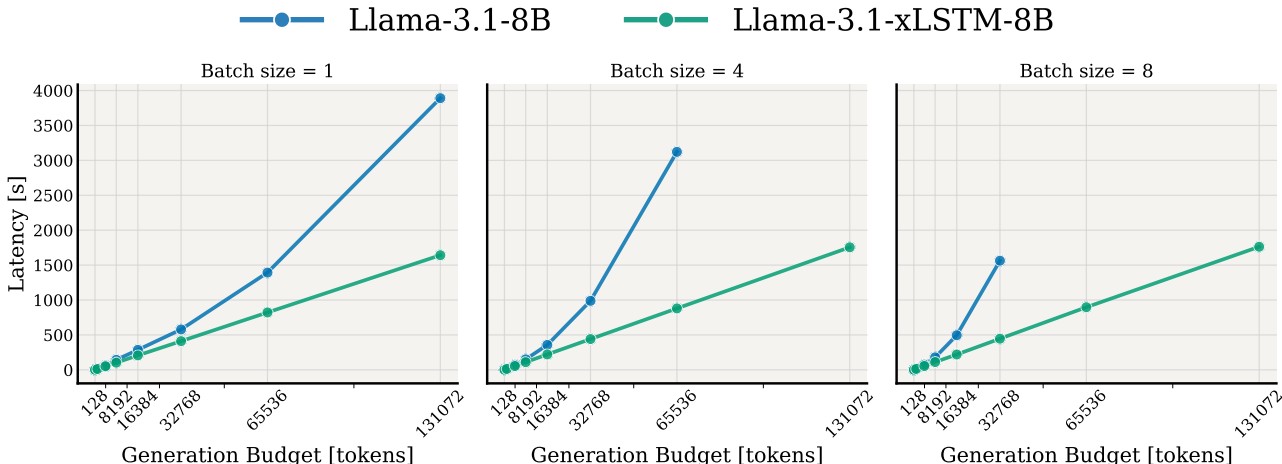

*Figure 14.* **Throughput**. We report the average throughput for generating 100 tokens with varying prefill lengths and batch sizes. Missing dots for the teacher indicate OOM.

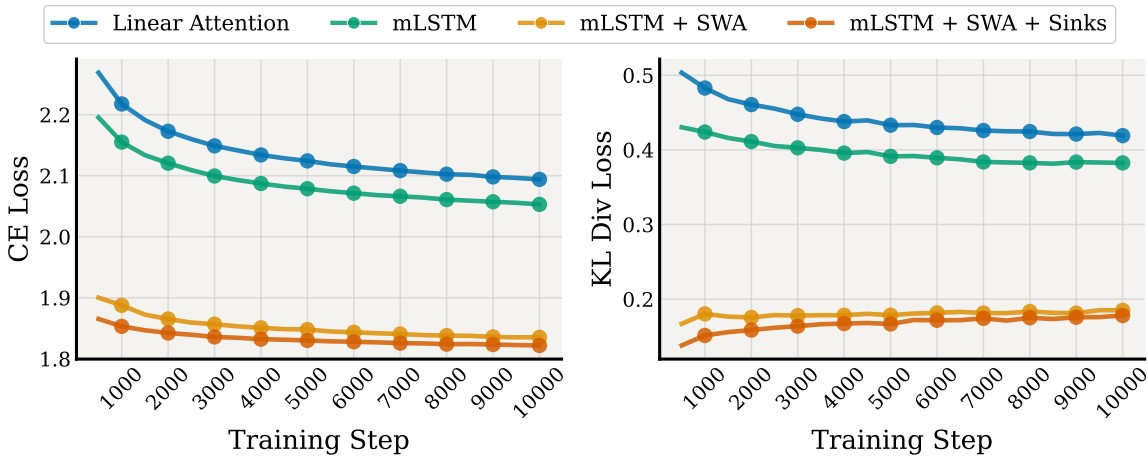

*Figure 15.* Ablation on the **effects of mLSTM, SWA & sinks** on our hybrid student. We track the cross-entropy loss (left) and KL loss (right) throughout stage II and across individual components. All components contribute considerably to the final performance

a window size $W = 512$ and designate the first four tokens of each sequence as attention sinks. We train with a CE/KL objective weighted by $\gamma = 0.9$ and $\beta = 0.1$. Evaluation loss curves in Figure 15 show that replacing linear attention with gated mLSTM reduces validation CE throughout training, indicating that head-wise gates increase expressivity and better match the teacher. Adding the SWA branch yields a further, uniform CE reduction, consistent with exact short-range recall. Introducing a small prefix of sink tokens provides an additional gain. Overall, the full hybrid (mLSTM + SWA + sinks) converges faster and achieves the lowest CE, i.e., the tightest student–teacher alignment.

### F.2. Effect of Distillation Objective

To find the best tradeoff between CE loss and the KL between the student and teacher models, we sweep over CE and KL loss weights $\gamma$ and $\beta$ (see Eq. 6). Evaluation losses of the different fine-tuning configurations are shown in Figure 16. As the KL $\beta$ grows, validation CE rises and the student under-adapts. As $\beta \to 0$, CE is lowest but the student drifts, with KL diverging. Prior work observes that large post-finetuning KL correlates with forgetting of capabilities (Shenfeld et al., 2025). Based on this, we adopt CE $\gamma$=0.9 and KL $\beta = 0.1$. This setting achieves CE essentially matching the $\gamma = 1.0, \beta = 0$ configuration while keeping KL dramatically smaller, providing substantial freedom to adapt to the new attention operators without sacrificing teacher alignment. Notably, even a small KL term materially improves alignment. Therefore, we use CE $\gamma = 0.9$ and KL $\beta = 0.1$ as our base setting.

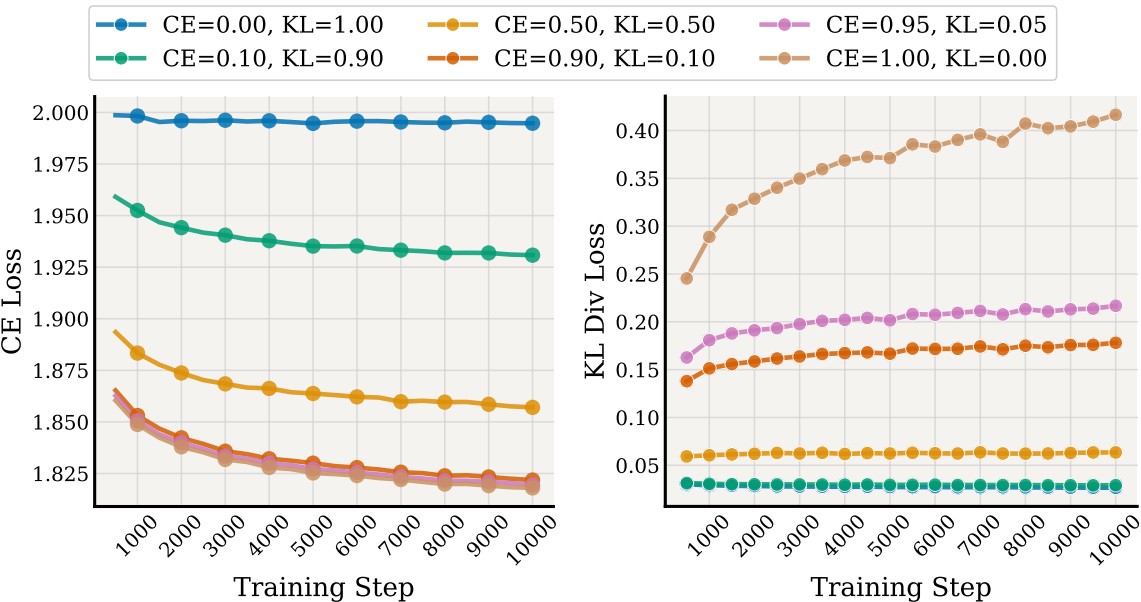

*Figure 16.* Ablation on the **effects of different loss weightings** on our hybrid student. We track the CE (left) and KL loss (right) throughout stage II. We find that weighting CE loss with 0.9 and KL loss with 0.1 provides a good tradeoff between performance and teacher alignment.

## F.3. PEFT vs. FFT

Prior linearization recipes use low-rank adaptation (LoRA, Hu et al., 2022) to recover performance lost during conversion (Zhang et al., 2025; Nguyen et al., 2025). While low-rank adaptation (LoRA) is attractive for its cost and scalability, it is unclear whether its capacity suffices to close the student-teacher gap. We therefore ablate three strategies spanning the efficiency–expressivity trade-off: (i) LoRA with high ranks ($r = 256$), (ii) updating only the sequence-mixer parameters while freezing all MLP blocks and embeddings, and (iii) FFT. We follow the baseline linearization setup in Section 3 and train with CE weight $\gamma = 0.9$ and KL weight $\beta = 0.1$. Figure 17 reports validation CE and KL. As expected, CE decreases as more parameters are unfrozen: FFT achieves the lowest CE, followed by mixer-only, then LoRA. Surprisingly, this additional flexibility does not increase deviation from the teacher. A small KL penalty is sufficient to keep all three methods comparably close in KL. Consequently, we adopt FFT, which offers the greatest capacity to adapt to the new attention operators while remaining close to the teacher model.

## F.4. Effect of Phases I&II

We find that layer-wise hidden-state alignment (Phase I) is necessary but not sufficient to recover most of the teacher's performance within a limited training budget. This finding aligns with previous work by (Zhang et al., 2025). Consequently, a two-phase approach, hidden-state alignment followed by full finetuning (Phase II), consistently outperforms standard finetuning given the same budget. While Phase I is crucial for aligning the student's intermediate hidden representations with the teacher's, the performance gains from this stage alone quickly plateau after a limited number of steps (Figure 18).

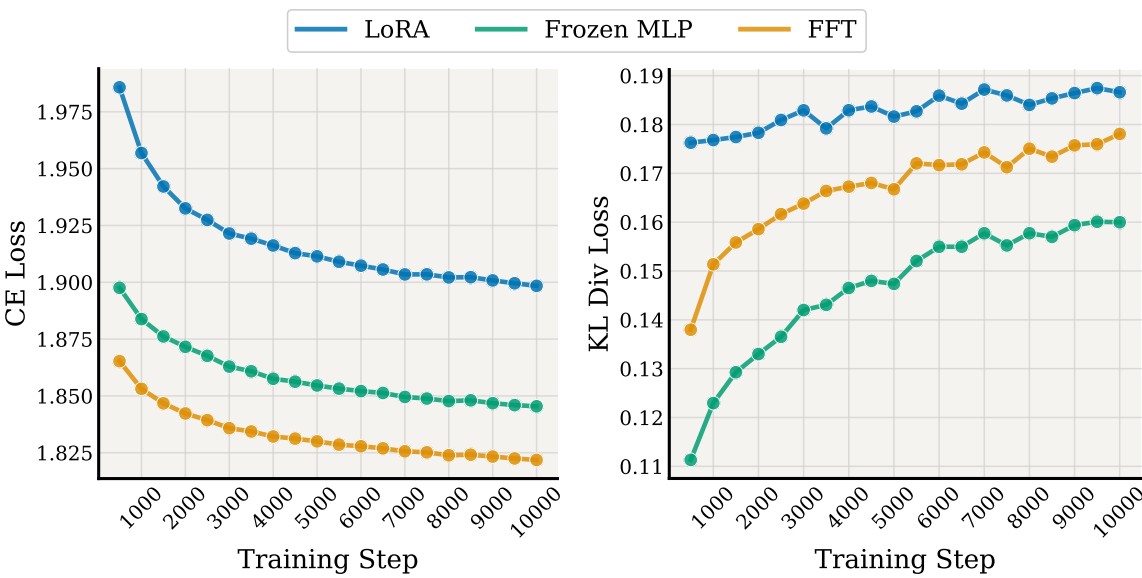

*Figure 17.* Ablation on the **effects of different training strategies** on our hybrid student. We track the CE loss (left) and KL loss (right) throughout stage II. This analysis reveals that full finetuning achieves a comparable KL while yielding superior downstream performance.

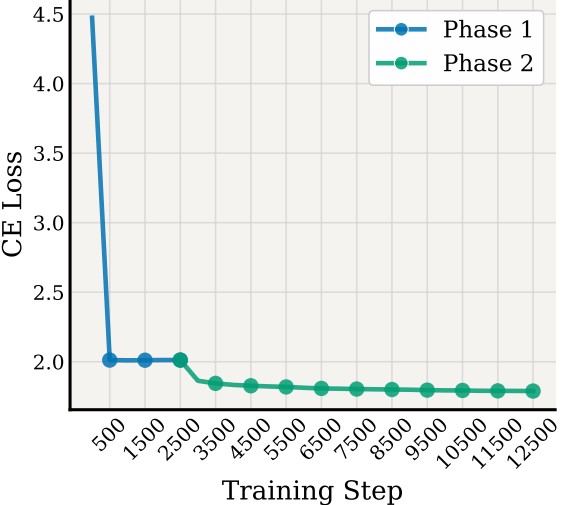

*Figure 18.* Phase I&II Ablation

