# OpenReview forum: "Effective Distillation to Hybrid xLSTM Architectures"
_ICML.cc/2026/Conference — ICML 2026 regular_

### Official Review · Reviewer_mTRh · 2026-03-05

**Soundness:** 2
**Presentation:** 3
**Significance:** 2
**Originality:** 2
**Overall Recommendation:** 3
**Confidence:** 3

**Summary:**

This paper focuses on distilling LLMs with quadratic attention mechanisms into efficient sub‑quadratic xLSTM‑based architectures. The core contribution is a complete distillation pipeline that includes: (1) a hybrid student architecture that integrates mLSTM, sliding‑window attention (SWA), and sink tokens through data‑dependent gating; (2) a two‑stage distillation process consisting of layer‑wise hidden‑state alignment and sparse knowledge distillation using top‑k KL divergence; (3) an optional expert merging stage, where domain‑specialized linearized models are trained independently and then combined via linear weight averaging; and (4) a formal evaluation metric, the tolerance‑corrected Win‑and‑Tie rate ($C_\alpha$), designed to evaluate whether the student model can serve as a reliable drop‑in replacement for the teacher model. The authors validate their method on multiple LLM families including Llama, Qwen, and OLMo, showing that their approach achieves much stronger performance on language understanding and challenging generative tasks than previous linearization methods such as LoLCATs, RADLADS, and Mamba‑in‑Llama, while also providing clear inference efficiency gains in throughput, latency, and memory usage.

**Compliance With Llm Reviewing Policy:**

Affirmed.

**Key Questions For Authors:**

See weakness

**Strengths And Weaknesses:**

## Strengths
1. Comprehensive and rigorous evaluation benchmark: The paper covers both language understanding and challenging generative tasks that are often neglected in prior work. The authors conduct thorough ablations on architectural components, loss functions, tuning strategies, and training stages. These experiments validate the contribution of each component and strengthen the technical soundness.
2. Well‑documented inference efficiency improvements: The paper provides detailed measurements of prefill throughput, generation latency, and GPU memory usage across various batch sizes and context lengths. It demonstrates consistent 2–4× real‑world speedups over highly optimized transformer teachers, independent of theoretical complexity claims.

## Weaknesses
1. Catastrophic long‑context retrieval failure: The student model suffers severe performance degradation on the NIAH task at 4K context length, with accuracy dropping to around 14.8%, while the teacher model maintains nearly perfect performance. This directly contradicts the core motivation of building efficient long‑context models, yet the paper provides insufficient analysis and no effective solution.
2. Flaws in the $C_\alpha$ evaluation metric and selective reporting: The $C_\alpha$ metric only considers the proportion of benchmarks where the student is not worse than the teacher, ignoring task importance and absolute performance gaps. The critical tolerance $\alpha^*$ masks severe drops on difficult tasks. Moreover, the paper prominently reports favorable generative results in the main text while relegating poor NIAH results to the appendix, raising concerns about selective reporting.
3. Primitive and poorly justified expert merging: The method uses simple linear weight averaging with heuristically chosen coefficients.  Merging also causes consistent performance degradation on STEM tasks across model families, but the authors do not adopt advanced merging methods such as TIES‑Merging or provide theoretical analysis.
4. Lack of theoretical analysis and limited generalization: The paper lacks rigorous theoretical guarantees, including expressivity bounds for the mLSTM‑SWA hybrid architecture, convergence guarantees for distillation, or performance bounds after weight merging.

---

> ### Author Rebuttal · Authors · 2026-03-30
>
> Thank you for the detailed review. We appreciate that you highlighted the comprehensive evaluation and the well-documented 2-4x efficiency gains.
>
> **Weakness #1. Long-context retrieval failure.** We agree the student underperforms on retrieval-heavy long-context benchmarks. However, prior work shows mLSTM+SWA hybrids can achieve substantially stronger long-context results [1], and PRISM demonstrates that short-context specialization degrades long-context retrieval even in strong base Transformers, with much of the loss recoverable via weight merging + long-context tuning [2]. Recent hybrid models (Kimi Linear, Nemotron 3) show strong long-context is compatible with partial attention retention [4-6]. We interpret our drop as a forgetting issue.
>
> PRISM observed the same pattern on Granite-3.3 8B: RULER at 128k collapsed from 59.09 to 6.46 after Math/Code mid-training, then recovered to 42.16 after merge, demonstrating this is not specific to our setup. To test this, we ran a control: Qwen3-4B-Instruct fine-tuned on Nemotron-Math-v2 [3]:
>
> | Benchmark | Base | After Math FT |
> |-|-|-|
> | AIME24 | 0.49 | 0.50 |
> | AIME25 | 0.40 | 0.45 |
> | HMMT0225 | 0.22 | 0.28 |
> | HMMT1125 | 0.30 | 0.32 |
>
> Notably, fine-tuning causes a severe collapse on MRCR Needle Task [4] for the Transformer (see table below). While the base model performs well, the fine-tuned checkpoint exhibits severe performance drops across different context lengths. Crucially, a simple 0.5/0.5 weight merge largely restores performance. This mirrors the observations from PRISM.
>
> | Model | 4k–8k | 8k–16k | 16k–32k | 32k–64k | 64k–128k |
> |-|-|-|-|-|-|
> | Base | 0.50 | 0.55 | 0.55 | 0.39 | 0.39 |
> | After Math FT | 0.28 | 0.08 | 0.04 | 0.02 | 0.01 |
> | 0.5 × Base + 0.5 × After Math FT | 0.56 | 0.54 | 0.49 | 0.39 | 0.27 |
>
> We therefore view long-context recovery as a separate post-specialization stage rather than a fundamental architectural barrier. Natural remedies include retaining a few attention layers, explicit long-context restoration training, and merging for capability patching. All of which fit naturally in our pipeline.
>
> **Weakness #2. Cα and selective reporting.** We respectfully think there is a misunderstanding about what Cα measures. Cα is a summary statistic and is intentionally designed to capture cross-benchmark consistency; how often the student matches or exceeds the teacher, rather than the magnitude of wins/losses. It is therefore complementary to average accuracy and per-benchmark reporting, which we also provide, not a replacement for them. A weighted variant is also possible if the community agrees on task weights. Nevertheless, we take the concern seriously: to avoid any impression of selectivity, we move NIAH into the main paper and discuss underperformance cases explicitly. This is also consistent with the broad empirical presentation that you and reviewer TgbD positively highlighted.
>
> **Weakness #3. Expert merging.** We did evaluate a modified TIES-Merging baseline [3]. Because our students replace attention weights with mLSTM/SWA parameters that have no direct base-model counterparts, TIES can only be applied to shared parameters; newly introduced parameters still require linear averaging. In initial 3-way merge experiments, this modified TIES did not outperform simple linear merging on average, so we did not pursue 4-way TIES further. Merge weights are tuned on held-out validation data, not reported test sets. Our findings are consistent with prior observations that simple weight averaging is often a surprisingly strong baseline [5]. We also note a practical constraint seen in production-scale merging pipelines: evaluation, not merging itself, is the real bottleneck in terms of compute [6]. We nevertheless agree that STEM degradation after merging is a limitation and will state this more clearly.
>
> **Weakness #4. Lack of theory / limited generalization.** We agree that the paper does not provide formal expressivity, convergence, or post-merge performance guarantees, and we will add this explicitly as a limitation. We respectfully disagree, however, that the absence of theory implies limited empirical generalization: the paper evaluates multiple teacher families, both base and instruction-tuned models, and a broad benchmark suite. Theoretical guarantees for modern heterogeneous LLM distillation pipelines, especially with a subsequent merging stage, remain largely open more generally.
>
> We believe these clarifications and additions substantially strengthen our submission, and we hope you will reconsider your assessment of our work's contribution. If any questions remain or come up during the rebuttal process, we would be happy to engage in further discussion.
>
> **References** \
> [1] https://arxiv.org/abs/2509.24552 \
> [2] https://arxiv.org/abs/2603.17074 \
> [3] https://arxiv.org/abs/2306.01708 \
> [4] https://huggingface.co/datasets/openai/mrcr \
> [5] https://arxiv.org/abs/2203.05482 \
> [6] https://arxiv.org/abs/2504.00698

---

> > ### Author Rebuttal · Reviewer_mTRh · 2026-04-07
> >
> > A theoretical analysis explaining why the hybrid architecture and merging strategy are effective should not be addressed only briefly in the rebuttal. I suggest the authors conduct a thorough revision of the paper accordingly.

---

> > > ### Author Response · Authors · 2026-04-07
> > >
> > > We appreciate the follow-up. We would like to stress that the aim of our paper is to propose a new distillation recipe and to validate the resulting models through broad experiments across multiple teacher families. A central motivation for this work is that prior linearization methods showed substantial gaps on challenging free-form generation tasks even when they performed well on language understanding benchmarks. Our paper identifies this gap clearly and substantially closes it, while also delivering clear inference gains. We believe these findings are useful to the linearization community.
> > >
> > > A full theoretical treatment of hybrid linearized architectures and model merging would be a substantial additional contribution in its own right and is beyond the scope of this paper. We agree that stronger theory in this area would be valuable. At the same time, these questions remain open more broadly, including in prominent related work on linearization and distillation [1,2,3,4]. We will make this limitation more explicit in the paper.
> > >
> > > [1] [https://arxiv.org/abs/2408.10189](https://arxiv.org/abs/2408.10189) \
> > > [2] [https://arxiv.org/abs/2408.15237](https://arxiv.org/abs/2408.15237) \
> > > [3] [https://arxiv.org/abs/2505.03005](https://arxiv.org/abs/2505.03005) \
> > > [4] [https://arxiv.org/abs/2410.10254](https://arxiv.org/abs/2410.10254)

---

### Official Review · Reviewer_TgbD · 2026-03-10

**Soundness:** 3
**Presentation:** 3
**Significance:** 2
**Originality:** 2
**Overall Recommendation:** 3
**Confidence:** 4

**Summary:**

The paper proposes a distillation pipeline that replaces transformer attention with a hybrid mLSTM–SWA linearized architecture, aiming to create efficient LLM students that closely match their teachers. It introduces a staged distillation process and an optional expert-merging strategy, along with a Win-and-Tie metric to evaluate teacher performance recovery across benchmarks. Experiments across Llama, Qwen, and Olmo models show strong performance recovery and improved inference efficiency compared to prior linearization methods. While the empirical results are promising and the problem is important, in my opinion the main contributions are mostly an integration of existing techniques, and some claims and evaluation choices may require clearer justification. Furthermore, no insight or intuition into why the approach works are presented.

**Compliance With Llm Reviewing Policy:**

Affirmed.

**Final Justification:**

think the main contribution here is merely an engineering combination of existing methods. Furthermore, the restriction to xLSTM makes the scope of the work extremely limited. During the rebuttal, the authors main arguments is their findings are extendable to other models, e.g, mamba2, RWKV-6. However, the current draft and experiments do not show or validate this claim, as they only focus xLSTM. I invite the authors, in their next submission, to try their approach with the aforementioned models as that can strengthen the paper and the contribution.

**Key Questions For Authors:**

- What is the key conceptual insight (novelty) beyond combining existing techniques into a pipeline?

- Which datasets or benchmarks were used to tune merge weights, and are they fully disjoint from the reported evaluation tasks?

- Were competing linearization baselines trained under comparable data and compute budgets?

- Beyond xLSTM, can the method be applied to other models such as Mamba or RWKV?

- How sensitive are the conclusions to the benchmark set $C_\alpha$ and the choice of tolerance threshold  $\alpha$?

**Limitations:**

yes

**Strengths And Weaknesses:**

**strengths:**

- Important and timely problem: Investigates whether linearized architectures can serve as practical replacements for transformer LLMs, a question with clear efficiency implications.
- Strong empirical scope: Distills multiple teacher families (Llama, Qwen, Olmo) across both base and instruction-tuned models, which strengthens the credibility of the results.

**Weaknesses:**
- Limited conceptual novelty: In my opinion, all the components (xLSTM, sparse attention, hidden-state alignment, KD, model merging) are known; the main contribution appears to be their integration rather than a fundamentally new method.

- Limited scope: the method is tailored for xLSTM, which limits the applicability of the proposed approach. It is unclear if it can generalize to other linear models, e.g.., mamba etc..

- The paper shows empirical gains but provides limited insight into why the hybrid architecture and merging strategy work.

- The term 'lossless distillation' is not accurate and is an overclaim. In fact, the empirical results show “near-lossless under tolerance” more than “lossless”.

---

> ### Author Rebuttal · Authors · 2026-03-30
>
> Thank you for the careful review. We appreciate your recognition that this is an important, timely problem and that the empirical scope across Llama, Qwen, and OLMo strengthens the study.
>
> **Weakness #1 / Question #1. Novelty beyond combining existing components.** Our contribution is identifying the recipe that closes the performance gap on free-form generation tasks, which prior work had not achieved at the same model scale. On our generation suite, LoLCATs and RADLADS require a high tolerance (α*=1.0) to match the teacher on half the benchmarks [1,2], Llamba relies on a 70B teacher [3], and Mamba-in-Llama retains 50% softmax attention and adds DPO [4]. In contrast, our pipeline achieves α*=0.0 for Llama-3.1-8B and α*=0.02/0.05 for instruction-tuned models. Beyond this, we contribute the Cα / α* evaluation framework and show that branch-train-merge remains effective after full linearization [5-7].
>
> **Weakness #2 / Question #4. Scope beyond xLSTM.** We agree that the current validation is specific to xLSTM, and we clarify that this paper does not claim cross-architecture validation. We chose mLSTM for our setup because its gated QKV formulation is natively compatible with Softmax Attention, its favorable scaling properties [8], and because recent evidence suggests higher wall-clock decoding throughput than Mamba-2 and GDN [9]. At the same time, we do not view the overall recipe as inherently limited to mLSTM: the main ingredients should transfer to other linear sequence mixers. In fact, reviewer R3gB explicitly noted that the approach likely generalizes to other linear-time architectures. We will elaborate this in the discussion section.
>
> **Weakness #3. Why the hybrid and merging work.** On merging, our claim is not a new theory of weight averaging, but that prior observations from model soups / branch-train-merge / TIES transfer to the fully linearized setting [5-7], which was not previously established. In this sense, we demonstrate that linearization can be made modular: linearized models can be combined via simple weight-space merging [6].
> On the architectural side, we already provide several empirical insights: our ablations isolate that mLSTM outperforms pure linear attention, SWA adds exact short-range recall, and sink tokens further increase the performance. The output gate analysis in Appendix E.2 (Figure 8) further reveals how the model learns to balance SWA and mLSTM across layers: mLSTM dominates early layers, SWA dominates middle layers, and later layers show a balanced mix. Together, these analyses provide insight into why the hybrid is more effective than either component alone.
>
> **Weakness #4 / Question #5. “Lossless distillation” and metric sensitivity.** We want to clarify that we do not claim to achieve lossless distillation. We formalize it as an evaluation target and use Cα and α* to quantify how close a student comes. Cα is a complementary summary, not a replacement for per-benchmark results. Our suite spans six domains, so Cα is not driven by any single task. Throughout the paper, we consistently report α* values and discuss results in terms of near-recovery rather than claiming parity. That said, we acknowledge the phrasing in the abstract ("We set out the goal of lossless distillation") could be read as an implicit claim. We will rephrase this to remove any ambiguity.
>
> **Question #2. Merge-weight tuning and data leakage.** We do not tune merge weights on reported test sets. For benchmarks with train/validation splits, we use held-out validation data (GSM8K, MATH, MBPP). Where no benchmark validation split exists, we use perplexity on held-out domain-specific distillation data plus simple heuristics. The strong performance on test-only benchmarks not used for tuning (e.g., HumanEval, CruxEval, GPQA, IFEval, MT-Bench) provides evidence that the chosen weights generalize rather than overfit. We will clarify the merge weight selection in more detail in the revised manuscript.
>
> **Question #3. Baseline compute/data comparability.** We agree that prior baselines are not trained under matched budgets, and we will add an explicit disclaimer. Importantly, the differences do not uniformly favor us: some baselines use much larger teachers (Llamba uses a 70B teacher for their 8B students, RADLADS uses a 72B teacher for 7B students) [2,3], and Mamba-in-Llama adds DPO and retains 50% of softmax attention [4].
>
> We hope these clarifications address your main concerns. We would be glad to discuss further and hope these additions are helpful for positively reassessing our work.
>
> **References** \
> [1] https://arxiv.org/abs/2410.10254 \
> [2] https://arxiv.org/abs/2505.03005 \
> [3] https://arxiv.org/abs/2502.14458 \
> [4] https://arxiv.org/abs/2408.15237 \
> [5] https://arxiv.org/abs/2208.03306 \
> [6] https://arxiv.org/abs/2203.05482 \
> [7] https://arxiv.org/abs/2306.01708 \
> [8] https://arxiv.org/abs/2510.02228 \
> [9] https://arxiv.org/abs/2512.15586

---

> > ### Author Rebuttal · Reviewer_TgbD · 2026-04-03
> >
> > Thank you for your response. However, I think the main contribution here is merely an engineering combination of existing methods. Furthermore, the restriction to xLSTM makes the scope of the work extremely limited. Hence, I stand with my original score.

---

> > > ### Author Response · Authors · 2026-04-03
> > >
> > > We thank the reviewer for the follow up.
> > >
> > > **Concern 1:** *Main contribution here is merely an engineering combination of existing methods*
> > >
> > > We respectfully disagree with the characterization of our work as “merely an engineering combination of existing methods.” The main contribution is a linearization recipe that materially narrows the teacher–student gap where prior work remains limited, particularly on free-form generation tasks. Rather than simply combining existing components, we make a series of targeted, linearization-specific architectural and training choices, and validate each of them empirically in this setting:
> > >
> > > - an initialization and gating design that leverages existing head-wise projections rather than only hidden states, while still collapsing back into an efficient inference form (as discussed in the paragraph starting at line 174).
> > > - a data-dependent convex output mixing formulation between the global recurrent path and local sliding-window path that, to our knowledge, has not appeared in prior linearization work in this form.
> > > - explicit retention of sink tokens, motivated by our finding that linearized models struggle to reproduce the sink behavior of the teacher, and supported by ablations showing that this is important for strong recovery. (Fig 7, Fig 15)
> > > - expert merging after full linearization, which is practically important because it makes it easier to balance the linearization token budget and enables decentralized capability training. Since modern RNNs/SSMs are known to be sensitive to numerical precision and recurrent-state perturbations [1], this is not obvious a priori.
> > > - the sparse top-k KD and a mixed CE/KL objective is novel in a linearization setting, with ablations supporting the chosen weighting (Fig 16). Top-k KD makes distillation substantially more efficient by enabling precomputation of teacher targets, and opening the door to distillation from much larger teacher models.
> > > - an open data recipe that goes beyond generic webtext data or small instruction-tuning sets, showing that data selection materially affects distillation quality for both base and instruction-tuned models.
> > > - The $C_\alpha$ / $\alpha^*$ evaluation framework, which complements per-benchmark scores by measuring how broadly teacher performance is recovered rather than only averaging gains.
> > >
> > > These architectural and recipe choices together lead to substantially stronger recovery than prior linearization baselines.
> > >
> > > **Concern 2:** *Restriction to xLSTM makes the scope of the work extremely limited*
> > >
> > > We also respectfully disagree that validating the recipe on xLSTM makes the scope “extremely limited.” Many modern linear-time sequence mixers share a closely related structure: Q/K/V-style projections, recurrent state updates, and learned gating or decay. In that sense, xLSTM is not an isolated special case, but one representative of a broader family of modern recurrent / linear-time architectures. This shared structure makes our recipe naturally compatible with other sequence mixers in the table below. We will revise the paper to make this cross-architecture compatibility more explicit.
> > >
> > > |Model|State update|Output|
> > > |-|-|-|
> > > |RetNet|$S_t = \gamma S_{t-1} + v_t k_t^\top$|$o_t = S_t q_t$|
> > > |GLA|$S_t = S_{t-1} \odot (\mathbf{1}\alpha_t^\top) + v_t k_t^\top$|$o_t = S_t q_t$|
> > > |Mamba-2|$S_t = \gamma_t S_{t-1} + v_t k_t^\top$|$o_t = S_t q_t$|
> > > |RWKV-6|$S_t = S_{t-1}\operatorname{Diag}(\alpha_t) + v_t k_t^\top$|$o_t = (S_{t-1} + (d \odot v_t)k_t^\top)q_t$|
> > > |Gated DeltaNet|$S_t = S_{t-1}\big(\alpha_t (I - \beta_t k_t k_t^\top)\big) + \beta_t v_t k_t^\top$|$o_t = S_t q_t$|
> > > |mLSTM|$S_t = f_t S_{t-1} + i_t v_t k_t^\top,\quad z_t = f_t z_{t-1} + i_t k_t$|$o_t = S_t q_t / \max(1, \lvert z_t^\top q_t \rvert)$|
> > >
> > > In post-training linearization, prior work typically evaluates one target sequence-mixer family at a time. For example, MOHAWK and Mamba-in-Llama focus on Mamba variants [2,3], RADLADS studies RWKV variants [4], and LoLCATs uses vanilla linear attention [5]. By that standard, evaluating one strong linear-time family is the norm, not an exceptional narrowing of scope. At the same time, our empirical scope within xLSTM is already broad: we study model sizes in the 7B to 8B range, multiple teacher families (Llama, Qwen, Olmo), and both base and instruction-tuned settings.
> > >
> > > We hope these clarifications address the reviewer's remaining concerns and motivate a reconsideration of the assessment of the paper's novelty and scope.
> > >
> > > [1] https://research.nvidia.com/labs/nemotron/files/NVIDIA-Nemotron-3-Super-Technical-Report.pdf \
> > > [2] https://arxiv.org/abs/2408.10189 \
> > > [3] https://arxiv.org/abs/2408.15237 \
> > > [4] https://arxiv.org/abs/2505.03005 \
> > > [5] https://arxiv.org/abs/2410.10254

---

### Official Review · Reviewer_R6eU · 2026-03-11

**Soundness:** 3
**Presentation:** 4
**Significance:** 3
**Originality:** 3
**Overall Recommendation:** 4
**Confidence:** 3

**Summary:**

This paper proposes an efficient distillation pipeline to convert Transformer-based LLMs into xLSTM-based linearized architectures, combining mLSTM and sliding window attention (SWA) to achieve strong performance recovery and significant inference efficiency gains. The method excels in language understanding and generation tasks but faces challenges in long-context modeling and domain interference during expert merging.

**Compliance With Llm Reviewing Policy:**

Affirmed.

**Final Justification:**

I have reviewed the rebuttal, and the author has addressed most of the concerns. I have already raised part of the score.

**Key Questions For Authors:**

How does the proposed hybrid attention mechanism perform on tasks requiring extreme long-context understanding (e.g., 100k tokens)? Can further optimizations like stronger memory designs help overcome these limitations?
How does the distillation pipeline scale with larger teacher models (e.g., GPT-3 or GPT-4-sized models)? Are there specific challenges in applying this method to sparse mixture-of-expert models?

**Limitations:**

The method has not been tested on larger models or sparse mixture-of-expert architectures, leaving its scalability and generalizability uncertain.

**Strengths And Weaknesses:**

Strengths
Efficiency Gains: The xLSTM-based student models significantly reduce the computational and memory costs during inference, achieving up to 4× faster generation throughput compared to Transformer-based teacher models.
Performance Recovery: The proposed pipeline enables student models to recover most of the teacher model's performance across language understanding and generation tasks, with some improvements in specific generation tasks like math reasoning and code synthesis.

Weaknesses
Long-Context Limitations: The student models underperform in long-context tasks such as the Needle-in-a-Haystack benchmark, where recall drops significantly as context length increases. This suggests limitations in the long-context modeling capabilities of the hybrid attention mechanism.
Limited Scale: The proposed method has only been tested on mid-sized models like LLAMA3.1-8B and QWEN2.5-7B. It remains unclear how well the pipeline scales to larger models or sparse mixture-of-expert architectures.

---

> ### Author Rebuttal · Authors · 2026-03-30
>
> Thank you for the constructive review and for highlighting both the efficiency gains and the strong recovery on language understanding and generation tasks.
>
> **Weakness #1 / Question #1. Long-context limitations and 100k+ contexts.** Our current interpretation is that the RULER/NIAH drop is primarily a forgetting issue induced by short-context specialization, consistent with PRISM [1]. PRISM observed this same pattern on Granite-3.3 8B: RULER at 128k collapsed from 59.09 to 6.46 after Math and Code mid-training, then recovered to 42.16 after weight merging with the base model and a short long-context tuning phase, demonstrating that this phenomenon is not specific to our setup. Prior work close to our setting shows that mLSTM+SWA hybrids can achieve substantially stronger long-context results, with performance depending strongly on training choices [2]. To verify whether this is specific to our converted hybrid or a broader post-training effect, we ran a preliminary control on a pure Transformer: Qwen3-4B-Instruct fine-tuned on Nemotron-Math-v2 [3]. This specialization improves in-domain math performance across all benchmarks:
>
> | Benchmark | Base | After Math FT |
> |-----------|------|---------------|
> | AIME24 | 0.49 | 0.50 |
> | AIME25 | 0.40 | 0.45 |
> | HMMT0225 | 0.22 | 0.28 |
> | HMMT1125 | 0.30 | 0.32 |
>
> Interestingly, the same specialization causes a severe collapse on MRCR Needle Task [4] (see table below). While the base model yields strong retrieval scores, the fine-tuned checkpoint shows severe performance degradation across different context lengths. Crucially, a simple 0.5/0.5 weight merge largely restores performance. This mirrors the observations from PRISM.
>
> | Model | 4k–8k | 8k–16k | 16k–32k | 32k–64k | 64k–128k |
> |-------|------|-------|--------|--------|---------|
> | Base | 0.50 | 0.55 | 0.55 | 0.39 | 0.39 |
> | After Math FT | 0.28 | 0.08 | 0.04 | 0.02 | 0.01 |
> | 0.5 × Base + 0.5 × After Math FT | 0.56 | 0.54 | 0.49 | 0.39 | 0.27 |
>
> We therefore view long-context recovery as a separate post-specialization stage rather than a fundamental architectural barrier. The most natural remedies are retaining a small number of attention layers, explicit long-context restoration training, and merging for capability patching. On the architectural side, increasing the mLSTM state dimension or exploring retrieval-augmented memory designs are promising complementary directions that we plan to investigate.
>
> **Weakness #2 / Questions #2-3. Scale to larger dense teachers and sparse MoE models.** We agree that the current empirical validation is limited to mid-sized dense models. Our goal for this work is to establish a practical distillation pipeline for linearizing strong pretrained LLMs. The experiments on multiple 7B–8B teacher families already provide meaningful evidence for the generality of the approach. That said, we do not see a method-specific reason why the pipeline would be restricted to this scale. The conversion primarily replaces softmax attention layers with mLSTM layers and does not appear to introduce a qualitatively new bottleneck beyond training the student itself. For larger dense teachers, the main constraint is therefore computational budget rather than a conceptual obstacle. For sparse MoE models, the additional challenges are mainly systems-related (especially routing and expert parallelism) rather than incompatibility with the recipe itself. We thus view extension to larger dense and sparse MoE models as an important next step, but not a prerequisite for validating the core contribution.
>
> We hope these clarifications and additions can further contribute to the positive assessment of our work and we would gladly engage in further discussions during the rebuttal.
>
> **References** \
> [1] https://arxiv.org/abs/2603.17074 \
> [2] https://arxiv.org/abs/2509.24552 \
> [3] https://huggingface.co/datasets/nvidia/Nemotron-Math-v2 \
> [4] https://huggingface.co/datasets/openai/mrcr

---

> > ### Author Rebuttal · Reviewer_R6eU · 2026-04-07
> >
> > The proposed method is intended for future application to large language models and Mixture-of-Experts (MoE) architectures, and its effectiveness has not yet been established.

---

> > > ### Author Response · Authors · 2026-04-08
> > >
> > > Thank you for the acknowledgement. We established the effectiveness of our method for efficient small- to medium-sized models (up to 8B), which are highly relevant in practice for local deployment, synthetic data generation, and specialized efficient use cases. We agree that validation on larger sparse MoE models remains future work. Such experiments are extremely costly and were not feasible within our compute budget.

---

### Official Review · Reviewer_R3gB · 2026-03-21

**Soundness:** 4
**Presentation:** 3
**Significance:** 4
**Originality:** 3
**Overall Recommendation:** 5
**Confidence:** 4

**Summary:**

This paper proposes a novel method to distill quadratic Transformer models into linear-time architectures. The training procedure consists of 3 stages: 1) training outputs of student’s sequence-mixing layer to match teacher’s quadratic attention layer via MSE loss; 2) classic distillation objective with soft and hard labels; 3) optional merging of several task-specific student models into one generalist model. It’s notable that the method is tightly coupled with architectural details: the authors use a gated combination of 1) SWA + attention sinks; and 2) linear-time global sequence-mixer module, mLSTM, slightly altered from its original design in https://arxiv.org/abs/2405.04517. The authors validate the performance and efficiency of the student models in comparison to the teacher models and obtain highly promising results.

**Compliance With Llm Reviewing Policy:**

Affirmed.

**Key Questions For Authors:**

1. lines 962-963: “We found that preserving the attention mask across these packed sequences, rather than truncating it, improved performance for our hybrid architecture”. I wasn’t able to grasp these lines. Could you explain how an attention mask should look in case of packed sequences?

2. line 1477: What do you mean by “pure linear attention” – the architecture from “Transformers are RNNs”, (https://arxiv.org/abs/2006.16236)?

3. Do you think the observed performance gains come due to your specific training recipe or underlying main architecture (mLSTM) or other architectural innovations on top (SWA + sinks)?

4. Regarding Appendix D – Did you use plain PyTorch or some framework like DeepSpeed or Lightning?


5. The previous question and many other questions that I have in my mind (e.g., does your implementation pf SWA attention have projection matrix $W_O$ besides $W_Q, W_K, W_V$) entails the following one: do you plan to open source the code? It would be nice to reproduce, adapt, and iterate on the results from your paper

6. Also, do you plan to open-source the models? This research benefited from the open-weights artifacts of previous works, such as LOLCATS, and it would be similarly beneficial for the community if you could release your distilled models.

**Limitations:**

Yes.

**Strengths And Weaknesses:**

**Strengths**

* The theme of this research is extremely important. There are many potent transformer-based models trained on tens of trillions of tokens, and unlocking their near lossless linearization would bring great benefits for inference speed, efficiency and even pre-training of next model generations.

* The empirical results are impressive. Based on data from Figures 3-4 and Tables 5-7, the student models match or even outperform the teacher models on most benchmarks. The paper demonstrates that 20B tokens and open-source datasets are enough to recover the performance of 7B models to a large degree.

* There are many evaluations and ablations which are helpful in understanding the performance of the method itself, and, importantly, could also be potentially applicable to distilling into other linear-time architectures (e.g. the effect of sink tokens).

* From the perspective of a researcher in the subfield of linear-time alternatives to Transformer, I find many ideas in the paper fresh, interesting and inspirational for application to my research. And it helps that the paper discloses many useful details which positively impacts adoption and further use of these ideas (e.g., specific token budgets, architecture and training hyperparemeters, etc.)

* I believe that the method isn’t tied only to mLSTM module but could be generalized to other linear-time architectures, current ones like Mamba-2 or GDA, and future ones. Moreover, I would like to see a follow-up work on this subject.

* I'm a strong proponent of making narration self-consistent, and, generally, this applies well to this paper (although I had to look up the original xLSTM paper to find out more about original mLSTM design).

**Weaknesses**

1. It’s puzzling for me that the student Llama-xLSTM greatly outperforms the teacher Llama-3.1 8B on GPQA-Diamond, GPQA, GSM8K, and HumanEval in Figure 3b, as well as other notably positive differences in Figures 4 and 9, and Tables 5-7. The outperformance of distilled xLSTM model in the range of 10-20% seems to be statistically significant, and it certainly necessitates a further investigation into the reasons why the student beats the teacher to such a great degree. I would appreciate a detailed and principled analysis of this phenomenon, and I believe, if such behavior indeed holds and is explained, it could become the most important finding of this paper.


2. You compare your approach with several other linearization methods like LOLCATS and RADLADS, which used different datasets, number of stages, number of tokens, and even training methods (full FT vs LoRA) for distillation. For example, RADLADS uses 700M tokens for their distillation of a comparable 7B model, while your procedure uses more than 20B. While I totally appreciate that it’s impossible to align and to bring their exact distillation procedures to a common denominator with your procedures, including data mix and token budget, it would be fair to put a disclaimer that all considered distillation procedures are not directly comparable, and the data in Tables 5-7 and Figures 3-4 does not necessarily indicate an absolute superiority of your method over counterparts.

3. Appendix A – the definition of lossless and $\alpha$-tolerant distillation is a little confusing. I would expect a distillation procedure to be $\alpha$-tolerant if the student model has better accuracy than $1 - \alpha$ level of the teacher model in *all benchmarks*, not just in half of them.

4. Small narrative mistakes:

* line 216 - found nowhere in the paper an explicit formula to calculate $o_t$

* You could also mention Kimi Linear https://arxiv.org/abs/2510.26692 around lines 840-841.

* lines 850-851: BASED architecture doesn’t interleave different mixers inside one layer, so it should be categorized as inter-layer hybrid (see Table 7 in https://openreview.net/forum?id=e93ffDcpH3 which implies that different attention mechanisms occupy separate layers).

---

> ### Author Rebuttal · Authors · 2026-03-30
>
> Thank you for the thoughtful review and supportive assessment. We especially appreciate your comments on the importance of the problem, the strong empirical results, the helpful ablations, and the potential applicability beyond mLSTM.
>
> **Weakness #1. Student outperformance over the teacher.** We agree this is interesting and deserves explanation. We attribute this mainly to data selection. The Llama-3.1-8B base student is distilled on Dolmino Mix [1], which Llama-3.1 was not originally trained on, whereas the OLMo-3-7B student (distilled on the already-used Dolmino 3 mid-training mix [2]) shows much less student-over-teacher gain. For instruction-tuned models, the experts are trained on Nemotron and other curated datasets [3]. Also, the largest relative gains (e.g., GPQA-D at 1.67x for Llama base) occur where teacher scores are low (10.6%), so modest absolute gains look larger under ratio-based recovery. Student-over-teacher gains under high-quality distillation data are also documented elsewhere [4,5].
>
> **Weakness #2. Baseline comparability.** We agree and will add an explicit disclaimer that cross-paper comparisons are not directly controlled for data mix, token budget, training strategy, or number of stages. Importantly, these differences rather favor other methods: Llamba uses up to 12B tokens and a much larger 70B teacher for their 8B students [6], RADLADS uses a 72B teacher for 7B students [7], and Mamba-in-Llama adds DPO and retains 50% softmax attention layers [8].
>
> **Weakness #3. “Lossless / α-tolerant” definition.** Our notion is performance equivalence, not uniform dominance. Requiring the student to stay within (1-α) of the teacher on all benchmarks would amount to uniform dominance, which is unrealistically strict. In contrast, Cα ≥ 0.5 captures that the student is not systematically worse across tasks. We will clarify this distinction more explicitly.
>
> **Weakness #4. Minor narrative issues.** We will make o_t explicit with a displayed equation (it's currently in the text around line 159), add Kimi Linear [9], and correct BASED to an inter-layer hybrid.
>
> **Question #1. Packed-sequence masks.** For softmax attention, the common alternative is a block-causal mask that also enforces document boundaries ([example](https://cdn-uploads.huggingface.co/production/uploads/624093c3ed76d6eadde83c48/8fDSe4Bebz9CdmUDiAQ-e.png) [10]). For mLSTM, the analogous operation is resetting recurrent state at document boundaries. We found better results with the standard causal mask and without truncating/resetting state across packed examples, consistent with recent findings on recurrent state reuse [11]. We will revise the paper to make this point more explicit and clarify that “preserving the attention mask” refers to not introducing additional document-boundary truncation or state reset beyond standard causal masking.
>
> **Question #2. “Pure linear attention.”** Here we remove the mLSTM input/forget gates, reducing it to a standard linear-attention operator with normalizer state, following the “Transformers are RNNs” view. For this ablation we use Hedgehog-style softmax feature maps. Empirically, these perform slightly better than standard choices such as ELU+1 or ReLU. We will clarify this.
>
> **Question #3. Recipe vs. architecture.** The gains come from both. Our ablations show mLSTM > pure linear attention, SWA and sink tokens each help, CE/KL weighting matters, full FT outperforms LoRA, and the two-stage pipeline is necessary. As you noted, these analyses also suggest transferability beyond mLSTM.
>
> **Question #4. Implementation.** We use vanilla PyTorch with FSDP2. We will clarify this in Appendix D.
>
> **Question #5. SWA projections / code release.** As in Figure 2, W_o is shared by the mLSTM and SWA branches: branch outputs are fused per head and then mixed through W_o. We will make this more explicit. Regarding code release, please refer to our response for question #6.
>
> **Question #6. Model release.** We plan to release the distilled models, and we are also working toward releasing the corresponding code. In parallel, we are integrating the intra-layer hybrid modules into common inference stacks such as vLLM. We expect this to unlock several promising directions, including reinforcement learning and on-policy distillation on top of our linearized models.
>
> If any points remain unaddressed, we would be very happy to engage in further discussion.
>
> **References** \
> [1] https://huggingface.co/datasets/allenai/dolmino-mix-1124 \
> [2] https://arxiv.org/abs/2512.13961 \
> [3] https://arxiv.org/abs/2508.14444 \
> [4] https://arxiv.org/abs/2308.02019 \
> [5] https://arxiv.org/abs/2502.08606 \
> [6] https://arxiv.org/abs/2502.14458 \
> [7] https://arxiv.org/abs/2505.03005 \
> [8] https://arxiv.org/abs/2408.15237 \
> [9] https://arxiv.org/abs/2510.26692 \
> [10] https://huggingface.co/blog/poedator/4d-masks \
> [11] https://arxiv.org/abs/2507.02782

---

> > ### Author Rebuttal · Reviewer_R3gB · 2026-04-04
> >
> > W1: Thank you for the links to papers documenting student models outperforming teachers! Please include your answer in the final revision.
> >
> > Q5, Q6: Please consider open-sourcing the code to reproduce your full distillation pipeline as well.
> >
> > Overall, I’m fully satisfied by the rebuttal, it’s clear and answers my questions precisely. I maintain a positive evaluation of this work and vote for its acceptance.

---

> > > ### Author Response · Authors · 2026-04-04
> > >
> > > Dear Reviewer R3gB,
> > >
> > > Thank you very much for your positive feedback on our rebuttal. We are very encouraged that our response addressed your main concerns.
> > >
> > > We will incorporate the references on student models outperforming their teachers in the revised version to make this point clearer in the paper. We also appreciate your suggestion regarding reproducibility and will release code and artifacts accompanying the paper as broadly as possible to support the community in reproducing our approach.
> > >
> > > Thank you again for your thoughtful feedback and for your support of the paper.

---

### Decision · Program_Chairs · 2026-04-30

**Decision:**

Accept (regular)

**Comment:**

This paper has mixed feedback during the review period. Reviewer R3gB and R6eU are leaning towards acceptance and believe the paper makes a solid contribution to the community, while Reviewer TgbD and mTRh have concerns.

On the positive side, there is broad agreement that the paper addresses an important problem and delivers strong empirical results. At the same time, Reviewer TgbD questioned the level of novelty, arguing that the method may be better understood as an effective integration of existing components rather than a fundamentally new technique. Reviewer mTRh also raised concerns about the lack of theoretical analysis.

Regarding novelty, I believe the paper tackles a new and important problem—distilling quadratic-attention LLMs into xLSTM-based architectures—and demonstrates strong empirical performance. In that sense, even though the method combines several previously proposed ingredients, the resulting contribution is still meaningful and sufficiently novel. Notably, all reviewers acknowledged the strength of the empirical results. I also think this work could open the door to future extensions that generalize the pipeline to a broader class of sub-quadratic LLM architectures.

Regarding theoretical analysis, I do not view the lack of a strong theoretical component as fatal in this case. In deep learning, theory is often difficult to formulate in a way that faithfully captures modern large-scale systems (especially LLMs), and many impactful papers are accepted primarily on the basis of strong empirical evidence. Here, the empirical results appear substantial enough to support the paper even without a deeper theoretical treatment.

Due to the importance of the problem and the strong experiments, I’m leaning towards acceptance. However, the author needs to revise the paper and incorporate the changes mentioned in the rebuttal (e.g., how to boost the long-context performance of the student models, how the method can be extended to other sub-quadratic LLM architectures).